# Probabilistic Modeling of Multi-rater Medical Image Segmentation for Diversity and Personalization

## Abstract

Medical image segmentation is inherently influenced by data uncertainty, arising from ambiguous boundaries in medical scans and inter-observer variability in diagnosis. To address this challenge, previous works formulated the multi-rater medical image segmentation task, where multiple experts provide separate annotations for each image. However, existing models are typically constrained to either generate diverse segmentation that lacks expert specificity or to produce personalized outputs that merely replicate individual annotators. We propose **Pro**babilistic modeling of multi-rater medical image **Seg**mentation (**ProSeg**) that simultaneously enables both diversification and personalization. Specifically, we introduce two latent variables to model expert annotation preferences and image boundary ambiguity. Their conditional probabilistic distributions are then obtained through variational inference, allowing segmentation outputs to be generated by sampling from these distributions. Extensive experiments on both the nasopharyngeal carcinoma dataset (NPC) and the lung nodule dataset (LIDC-IDRI) demonstrate that our ProSeg achieves a new state-of-the-art performance, providing segmentation results that are both diverse and expert-personalized.

## 1 Introduction

Medical image segmentation is of great importance for automatic diagnosis and treatment planning in clinical practice Isensee et al. (2021). However, the task is challenging due to the inherent uncertainty of data, such as medical scans, ambiguous boundaries Carass et al. (2017); Wu et al. (2023) and irregular shapes of medical targets Marin et al. (2022); Luo et al. (2023); Li et al. (2017); Fu et al. (2020), as well as the inter-observer variability in diagnosis Menze et al. (2014). To tackle this issue, the multi-rater medical image segmentation task was proposed to take the data uncertainty into account by collecting annotations from different experts for each image independently Rahman et al. (2023).

Existing approaches for multi-rater medical image segmentation are limited to either generating diverse segmentation that cannot resemble realistic expert variability (Kohl et al., 2018; Rahman et al., 2023) or individual-specific outputs simply mirroring annotators

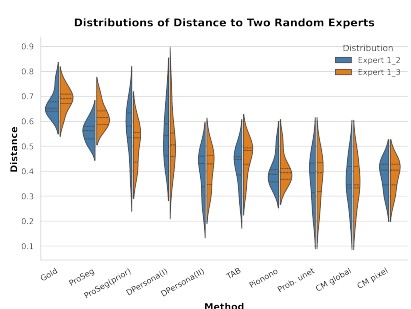

Figure 1: Distance distribution between two random experts. A greater distance indicates higher diversity and a more similar distribution with the Gold standard indicates better personalization.

(Liao et al., 2023; Schmidt et al., 2023). We calculated the distance between two random experts of different methods in **Fig. 1**, where the greater distance indicates higher segmentation diversity, while similarity to the gold standard suggests increased personalization. Specifically, *Generation methods*, like Probabilistic U-Net (Kohl et al., 2018), produce diverse and reliable segmentation results, yet fail to capture expert personalization (Rahman et al., 2023). *personalization methods*, like TAB (Liao et al., 2023), Pionono (Marin et al., 2022), CM global (Tanno et al., 2019), and CM Pixel (Zhang et al., 2020), which can replicate individual expert annotations, yet show lim-

ited diversity. *Crowdsourcing approaches* aggregate multiple annotations into a meta-segmentation, which is restricted by the assumption of one single meta-segmentation (Warfield et al., 2004). To bridge the gap between diversity and personalization, Wu et al. (2024) employed a two-stage DPersona to produce diverse segmentation initially and personalized ones subsequently, yet its efficacy is constrained by lacking probabilistic modeling. Thus, a unified probabilistic approach capable of simultaneously generating both diverse and personalized segmentation remains an open challenge.

To address the challenge of modeling multi-rater medical image segmentation considering both diversity and personalization, we propose **ProSeg**, a **Pro**bability graph model for multi-rater **Seg**mentation. Specifically, we introduce two latent variables $\tau$ and $Z$ to model the inter-observer variability in diagnosis and the ambiguity in medical scans, respectively, as shown in **Fig. 2(d)**, where $\tau$ and $Z$ are inferred from the observed data and annotations. By sampling from the latent space, we can generate diverse segmentation results, while specifying a particular expert allows us to produce personalized segmentation outputs that align with individual expert annotations. The model is trained by maximizing the evidence lower bound (ELBO), which consists of the log-likelihood of the observed data and annotations, as well as the regularization term ensuring similarity between prior and posterior distribution of $\tau$ and $Z$.

To demonstrate the effectiveness of ProSeg, we conducted extensive experiments on two benchmark datasets: the nasopharyngeal carcinoma (NPC) dataset and the lung nodule dataset (LIDC-IDRI). The empirical results indicate that ProSeg consistently outperforms previous methods, producing both diverse and expert-personalized segmentation, achieving state-of-the-art performance in multi-rater medical image segmentation. Furthermore, ProSeg serves as a generalizable framework that can be readily extended to other medical image segmentation tasks, highlighting its versatility and broad applicability.

To the best of our knowledge, ProSeg is the first probabilistic modeling framework that can simultaneously generate diverse and personalized segmentation results for multi-rater medical image segmentation. The main contributions of our work can be summarized as follows:

- We propose a unified probabilistic modeling framework, ProSeg, for multi-rater medical image segmentation, which generates both diverse and personalized segmentation results.

- We introduce two latent variables, $\tau$ and $Z$, via variational inference, to model the inter-observer variability in diagnosis and the ambiguity in medical scans, respectively.

- We conduct extensive experiments, demonstrating that ProSeg achieves a new state-of-the-art performance in multi-rater medical image segmentation.

## 2 RELATED WORKS

### 2.1 MULTI-RATER MEDICAL IMAGE SEGMENTATION

Multi-rater medical image segmentation aims to take the data uncertainty, including the inter-observer variability in diagnosis and the ambiguity in medical scans, into consideration. Existing methods can be broadly categorized into three groups: **crowdsourcing methods** combine multiple annotations to approach a meta-segment Warfield et al. (2004); **generation methods** learns a latent distribution to generate diverse segmentation Kohl et al. (2018); Rahman et al. (2023); and **personalization methods** produce personalized segmentation that aligns with individual expert annotations Zhang et al. (2020); Liao et al. (2023); Schmidt et al. (2023). Diversity and personalization are essential aspects of this task, ensuring that segmentation results capture variations across experts while aligning with individual annotations. However, these methods are generally limited to generating diverse segmentations that lack expert specificity or producing personalized outputs that merely replicate individual annotators.

To bridge this gap, Wu et al. (2024) attempted to train a two-stage model to generate diverse segmentation results in the first stage and personalized results in the second stage. However, its effectiveness is limited by the absence of a probabilistic modeling framework. To address these challenges, we introduce ProSeg, a probabilistic modeling framework capable of simultaneously generating both diverse and personalized segmentation results, offering a unified solution for multi-rater medical image segmentation.

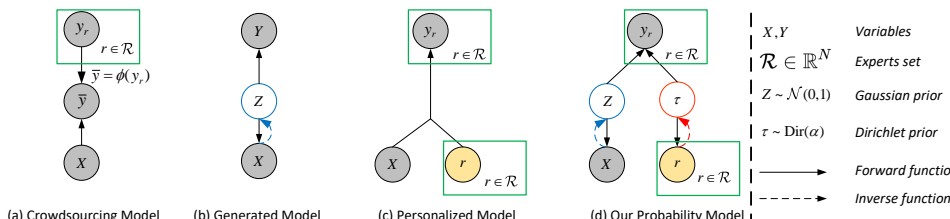

(a) Crowdsourcing Model     (b) Generated Model     (c) Personalized Model     (d) Our Probability Model

Figure 2: Probability graph model (PGM) of methods for multi-rater segmentation. $X$, $\mathcal{R}$, and $Y$ denote the images, expert annotators, and annotations respectively. The latent variable $Z$ denotes the ambiguity in medical scans. In our probability model, a latent variable $\tau$ is formulated to model the subjective variants among expert annotators. The green rectangular box represents a set of variables.

## 2.2 PROBABILISTIC MODELING

Probabilistic modeling has been extensively applied across various machine learning domains, including generative modeling (Kingma, 2013; Rezende et al., 2014), computer vision (Gao & Zhuang, 2022), and Bayesian optimization (Shahriari et al., 2015). By explicitly modeling uncertainty, probabilistic approaches provide robust predictions and facilitate better generalization, particularly in tasks where data is noisy or ambiguous. Existing probabilistic methods for medical image segmentation focus mainly on modeling uncertainty in medical scans, leading to improvements in model robustness and interpretability (Hatamizadeh et al., 2022; Wu et al., 2022; Gao et al., 2023). For example, Bayesian deep learning (Gal & Ghahramani, 2016) and Monte Carlo sampling (Kohl et al., 2018) have been used to estimate segmentation uncertainty. However, they do not address the combined challenges of diversity and personalization in multi-rater medical image segmentation. To our knowledge, no existing work has leveraged probabilistic modeling to simultaneously generate both diverse and personalized segmentation results for multi-rater medical image segmentation.

## 3 PRELIMINARIES AND NOTATION

### 3.1 THE MULTI-RATER MEDICAL IMAGE SEGMENTATION TASK

In the setting of multi-rater medical image segmentation, each medical image $\boldsymbol{x} \in \mathbb{R}^d$ is annotated by a group of expert annotators $\mathcal{R} = \{r_1, r_2, \ldots, r_N\} \in \mathbb{R}^N$, providing independent annotations $\{\boldsymbol{y}_r\}_{r \in \mathcal{R}}$, where, $\boldsymbol{y}_r \in \mathbb{R}^{dK}$ denotes the annotation provided by the expert annotator $r$, and $d$ and $K$ denote the image size and the number of segmentation classes, respectively. For simplicity, we define $d$ as the product of the image height and width, representing the image size in our notation. Therefore, for each image $\boldsymbol{x}$, we have the ensemble of its annotations, $Y$, where $Y = (\boldsymbol{y}_{r_1}, \boldsymbol{y}_{r_2}, \ldots, \boldsymbol{y}_{r_N}) \in \mathbb{R}^{dK \times N}$ denotes $N$ annotations from $\mathcal{R}$ respectively. We denote the dataset as $\mathcal{D} = \{\mathcal{R}^{(i)}, Y^{(i)}, \boldsymbol{x}^{(i)}\}_{i=1}^{|\mathcal{D}|}$, where $|\mathcal{D}|$ denotes the number of samples in the dataset. Thus, the multi-rater medical image segmentation aims to learn a mapping from the inputs $\boldsymbol{x}$ and $\mathcal{R}$ to the output segmentation $Y$. We can define the multi-rater medical image segmentation task as follows:

**Definition 3.1** (Multi-rater medical image segmentation). Given a dataset $\mathcal{D} = \{\mathcal{R}^{(i)}, Y^{(i)}, \boldsymbol{x}^{(i)}\}_{i=1}^{|\mathcal{D}|}$ of expert annotations and medical images, the multi-rater medical image segmentation task aims to learn the mapping from the input image $\boldsymbol{x}^{(i)}$ and $\mathcal{R}^{(i)}$ to the ensemble of multiple segmentation $Y^{(i)}$, *i.e.*, $Y^{(i)} = f(\boldsymbol{x}^{(i)}, \mathcal{R}^{(i)})$.

Previous works including crowdsourcing methods Warfield et al. (2004), generation methods Kohl et al. (2018); Rahman et al. (2023), and personalization methods Zhang et al. (2020); Liao et al. (2023) have been proposed to address the multi-rater medical image segmentation task. Crowdsourcing methods assume that a single meta-segmentation exists, which can be obtained by combining multiple annotations. They aggregate the annotations from different experts $Y^{(i)} \in \mathbb{R}^{dK \times N}$ into one meta-segment $\overline{y}^{(i)} \in \mathbb{R}^{dK}$ with various methods $\overline{y}^{(i)} = \phi(Y^{(i)})$, like majority voting or STAPLE Warfield et al. (2004) as shown in **Fig. 2 (a)**. We can remark crowdsourcing methods as:

*Remark* 3.2 (Crowdsourcing method). The crowdsourcing method aggregates the annotations from different experts into one meta-segment with various methods $\overline{y}^{(i)} = \phi(Y^{(i)})$. Then the meta-segment $\overline{y}^{(i)}$ is used as the ground truth for training a single segmentation model, *i.e.*, $\overline{y}^{(i)} = f(\boldsymbol{x}^{(i)})$.

Since the expert annotators are not included in the training process, the crowdsourcing method is unable to generate either diverse or personalized segmentation results.

Generation methods aim to generate diverse segmentation results (Kohl et al., 2018). They simultaneously learn a latent space $z = \psi(x)$ to make the conditional distributions $p(z|x)$ and $p(z|x, Y)$ identical as well as the mapping from the latent representation $z$ and the image $x$ to the segmentation results as shown in **Fig. 2(b)**. Then diversified segmentation results are generated by sampling from the latent space $z$. We can remark generation methods as:

*Remark* 3.3 (Generation method). Generation method learns a latent representation $z$ as well as the mapping from the image $x$ and latent representation $z$ to segmentation results, *i.e.*, $Y = f(x, z)$.

Since generation methods do not consider the expert annotators, they are struggling to generate personalized segmentation results.

Personalization methods aim to generate personalized segmentation results mimicking the corresponding expert annotators. They learn a conditional segmentation by incorporating the expert annotators into the training process as shown in **Fig. 2(c)**. We can remark the personalization method as:

*Remark* 3.4 (Personalization method). Personalization method learns a conditional segmentation by incorporating the expert annotators into the training process, *i.e.*, $y_r = f_r(x)$, $r \in \mathcal{R}$.

However, personalization methods typically hard-code expert annotators, restricting them to generating personalized segmentation results that merely replicate individual annotations, without producing diverse segmentation. Moreover, these methods aim to learn a one-to-one correspondence between the input image $x$ and the expert annotation $y_r$, ignoring variations in the annotator's personal preferences.

Due to the limitations of existing methods either generating diverse segmentation that lacks expert specificity or producing personalized outputs that merely replicate individual annotators. We take the variability of expert annotators and the uncertainty of ambiguous boundaries into account as shown in **Fig. 2(d)**. We reformulate multi-rater medical image segmentation as a probabilistic modeling problem that captures the joint distributions of experts, annotations, and medical images as follows:

**Definition 3.5** (Probabilistic modeling of multi-rater medical image segmentation). Probabilistic modeling of multi-rater medical image segmentation aims to model the joint distribution of experts, annotations, and medical images, *i.e.*, $p(Y, x, \mathcal{R})$.

Then the diversity and personalization can be defined as follows:

**Definition 3.6** (Diversity). Diversity in multi-rater medical image segmentation refers to the dissimilarity among samples from the distribution of segmentation given the image, *i.e.*, $p(\hat{Y}|x)$, where $\hat{Y}$ indicates the generated segmentation.

**Definition 3.7** (Personalization). Personalization in multi-rater medical image segmentation refers to the consistency between the ground truth $p(y_r|X, r)$ and predicted segmentation $p(\hat{y}_r|X, r)$ given the image and the specific annotation from the expert $r$.

## 4 METHODOLOGY

To model the joint distribution of experts, annotations, and medical images $p(Y, x, \mathcal{R})$, we propose a probabilistic modeling framework, ProSeg, that can generate both diverse and personalized segmentation results for multi-rater medical image segmentation. Generally, ProSeg introduces two latent variables, $\tau = (\tau_1, \tau_2, \ldots, \tau_N)$ and $Z = (z_1, z_2, \ldots, z_N)$, to model the subjective variations among expert annotators and the ambiguity in medical scans, respectively. The conditional probabilistic distributions of both variables are obtained through variational inference.

### 4.1 PROBABILISTIC MODELING OF MULTI-RATER MEDICAL IMAGE SEGMENTATION

We model the joint distribution of experts, annotations, and medical images $p(Y, x, \mathcal{R})$ as a probabilistic graphical model (PGM) as shown in Fig. 2(d). The PGM consists of two parts: the observed expert annotators $\mathcal{R}$, annotations $Y$, and medical images $X$, as well as the latent variables $\tau$ and $Z$. Since one image corresponds to multiple expert annotations, we introduce the latent variable $\tau$

Figure 3: Model architecture of deep variational inference for multi-rater segmentation. ProSeg consists of image decoders $p(\boldsymbol{x}|\boldsymbol{z}_i)$, image encoders $p(\boldsymbol{z}_i|\boldsymbol{x})$, class embedding $q(\tau|\mathcal{R})$, classifier $p(\mathcal{R}|\tau)$, and the segmentation predictor $p(\mathbf{y}_{r_i}|\tau_i, z_i)$.

to model the subjective variations among expert annotators, which is only related to the annotations $Y$ and the expert annotators $\mathcal{R}$. We also introduce the latent variable $Z$ to model the ambiguity in medical scans, which is only related to the medical images $\boldsymbol{x}$ and the annotations $Y$. The joint distribution $p(Y, \boldsymbol{x}, \mathcal{R}, \tau, Z)$ can be denoted as follows:

$$p(Y, \boldsymbol{x}, \mathcal{R}, \tau, Z) = p(Y, \boldsymbol{x}, \mathcal{R}|\tau, Z)p(\tau, Z). \tag{1}$$

The joint distribution $p(Y, \boldsymbol{x}, \mathcal{R})$ can be derived as the marginalization over the two latent variables, $\tau$ and $Z$, as follows:

$$p(Y, \boldsymbol{x}, \mathcal{R}) = \iint p(Y, \boldsymbol{x}, \mathcal{R}|\tau, Z)p(\tau, Z)d\tau dZ. \tag{2}$$

According to the chain rule of probability, we can factorize the conditional distribution $p(Y, \boldsymbol{x}, \mathcal{R}|\tau, Z)$ into two parts as follows:

$$p(Y, \boldsymbol{x}, \mathcal{R}|\tau, Z) = p(Y|\boldsymbol{x}, \mathcal{R}, \tau, Z)p(\boldsymbol{x}, \mathcal{R}|\tau, Z). \tag{3}$$

For the first part $p(Y|\boldsymbol{x}, \mathcal{R}, \tau, Z)$, since the annotations $Y$ are only related to the latent variables $\tau$ and $Z$ in our modeling, we can simplify the conditional distribution $p(Y|\boldsymbol{x}, \mathcal{R}, \tau, Z)$ as $p(Y|\tau, Z)$. Given the annotators $\mathcal{R}$ are independently annotated images, we can denote the conditional distribution $p(Y|\tau, Z)$ as the product of $p(\mathbf{y}_{r_i}|\tau_i, \boldsymbol{z}_i)$ as follows:

$$p(Y|\tau, Z) = \prod_{i=1}^{N} p(\mathbf{y}_{r_i}|\tau_i, \boldsymbol{z}_i). \tag{4}$$

More details can be found in the Appendix. B.2.

For the second part, $p(\boldsymbol{x}, \mathcal{R}|\tau, Z)$, we can further factorize the conditional distribution according to the chain rule of probability as follows:

$$p(\boldsymbol{x}, \mathcal{R}|\tau, Z) = p(\boldsymbol{x}|\mathcal{R}, \tau, Z)p(\mathcal{R}|\tau, Z). \tag{5}$$

Since the medical image $\boldsymbol{x}$ is only related to the latent variables $Z$ in our modeling, we can simplify the conditional distribution $p(\boldsymbol{x}|\mathcal{R}, \tau, Z)$ as $p(\boldsymbol{x}|Z)$. We further express $p(\boldsymbol{x}|Z)$ as the product of the conditional distributions, $p(\boldsymbol{x}|\boldsymbol{z}_i)$, since one image corresponds to multiple understandings of experts and thus multiple independent annotations. Similarly, we can simplify the conditional distribution $p(\mathcal{R}|\tau, Z)$ as $p(\mathcal{R}|\tau)$, since the expert annotators $\mathcal{R}$ are only related to the latent variable $\tau$ in our modeling. Due to the independency nature of expert annotators, we can express $p(\mathcal{R}|\tau)$ as the product of the conditional distributions, $p(r_i|\tau_i)$. Then the conditional distribution $p(\boldsymbol{x}, \mathcal{R}|\tau, Z)$ can be converted to:

$$p(\boldsymbol{x}, \mathcal{R}|\tau, Z) = \prod_{i=1}^{N} p(\boldsymbol{x}|\boldsymbol{z}_i)p(r_i|\tau_i). \tag{6}$$

We have unfolded the generation process as defined in Eq. 3 by incorporating the generative distributions $p(\boldsymbol{x}|\boldsymbol{z}_i)$, $p(\mathbf{y}_{r_i}|\boldsymbol{z}_i)$, and $p(r_i|\tau_i)$. Inversely, we model the variational distribution of $\boldsymbol{z}_i$ given $\boldsymbol{x}$, denoted as $q(\boldsymbol{z}_i|\boldsymbol{x})$, and the variational distribution of $\tau$ given $\mathcal{R}$, denoted as $q(\tau|\mathcal{R})$, to approximate the posterior distributions of $\boldsymbol{z}_i$ given $\boldsymbol{x}$ and $\tau$ given $\mathcal{R}$, respectively. Since the variational

distributions $q(\boldsymbol{z}_i|\boldsymbol{x})$ and $q(\tau|\mathcal{R})$ are unknown, we can estimate them by the variational inference. To achieve that, we introduce a Gaussian prior for $Z$, *i.e.*, $p(Z) = \prod_{i=1}^{N} p(\boldsymbol{z}_i) = \prod_{i=1}^{N} \mathcal{N}(0, I)$, and a Dirichlet prior for $\tau$, *i.e.*, $p(\tau) = \prod_{i=1}^{N} p(\tau_i) = \prod_{i=1}^{N} \mathrm{Dir}(\alpha_0)$, where $\alpha_0 = \mathbf{1} \in \mathbb{R}^N$ is the concentration hyperparameter with each element to be one.

Finally, to learn the joint distribution $p(Y, \boldsymbol{x}, \mathcal{R})$, we can maximize the ELBO of the evidence, $\ln p(Y, \boldsymbol{x}, \mathcal{R})$, which is equivalent to minimizing the negative observation log-likelihood as well as the KL divergence between the prior and posterior distributions of $Z$ and $\tau$ as follows:

$$-\mathbb{E}_{q(Z|\boldsymbol{x}), q(\tau|\mathcal{R})} \left[ \ln p(Y, \boldsymbol{x}, \mathcal{R}|Z, \tau) \right] + \mathrm{KL}\left( q(Z|\boldsymbol{x}) || p(Z) \right) + \mathrm{KL}\left( q(\tau|\mathcal{R}) || p(\tau) \right). \quad (7)$$

## 4.2 MODEL ARCHITECTURE

With the probabilistic modeling of multi-rater medical image segmentation above, we refactor the problem of learning the joint distribution $p(Y, \boldsymbol{x}, \mathcal{R})$ into learning the distribution of $p(\boldsymbol{x}|\boldsymbol{z}_i)$, $p(\boldsymbol{y}_{r_i}|\tau_i, \boldsymbol{z}_i)$, $p(\mathcal{R}|\tau)$, $q(\boldsymbol{z}_i|\boldsymbol{x})$, and $q(\tau|\mathcal{R})$. We model these probabilities with neural networks as shown in Fig. 3. The model consists of five modules, including the image encoder, image decoder, class embedding, classifier, and segmentation predictor. We infer the distributions $p(\boldsymbol{x}|\boldsymbol{z}_i)$ and $q(\boldsymbol{z}_i|\boldsymbol{x})$ with the image decoders $p_{\theta_i}(\boldsymbol{x}|\boldsymbol{z}_i)$ and image encoders $q_{\phi_i}(\boldsymbol{z}_i|\boldsymbol{x})$. The distributions $p(\mathcal{R}|\tau)$ and $q(\tau|\mathcal{R})$ are learned with the class embedding $q_{\phi_\tau}(\tau|\mathcal{R})$ and classifier $p_{\theta_\tau}(\mathcal{R}|\tau)$. The distribution $p(\boldsymbol{y}_{r_i}|\tau_i, \boldsymbol{z}_i)$ is estimated with the segmentation predictor $p_{\theta_Y}(\boldsymbol{y}_{r_i}|\tau_i, \boldsymbol{z}_i)$. It is worth noting that we use a single segmentation predictor instead of multiple ones.

To train the neural networks, we minimize the loss in equation 7. For the negative log-likelihood (NLE), we can denote the loss as the reconstruction loss of image, the classification of expert annotators, and the segmentation loss as follows:

$$\mathcal{L}_{\mathrm{NLE}}(\Phi, \Theta; Y, \boldsymbol{x}, \mathcal{R}) = \mathcal{L}_{\mathrm{recon}} + \mathcal{L}_{\mathrm{class}} + \mathcal{L}_{\mathrm{seg}} = \mathbb{E}\left[ d(\hat{\boldsymbol{x}}, \boldsymbol{x}) \right] + \mathbb{E}\left[ \mathcal{R} \log \hat{\mathcal{R}} \right] + \mathbb{E}\left[ Y \log \hat{Y} \right], \quad (8)$$

where, $\Phi = \{\phi_1, \ldots, \phi_N, \phi_\tau\}$, $\Theta = \{\theta_1, \ldots, \theta_N, \theta_\tau, \theta_Y\}$, $\hat{\boldsymbol{x}}$, $\hat{\mathcal{R}}$, and $\hat{Y}$ are the prediction of images, expert annotators, and annotations, respectively, and the expectations are over the variational distributions $q_{\phi_i}(\boldsymbol{z}_i|\boldsymbol{x})$ and $q_{\phi_\tau}(\tau|\mathcal{R})$. In the reconstruction loss $\mathcal{L}_{\mathrm{recon}}$, $d(\cdot, \cdot)$ denotes the mean squared error between the predicted image and the ground truth image. The classification loss $\mathcal{L}_{\mathrm{class}}$ is the cross-entropy loss between the predicted expert annotators and the ground truth expert annotators. The segmentation loss $\mathcal{L}_{\mathrm{seg}}$ is the cross-entropy loss between the predicted annotations and the ground truth annotations. More details can be found in the Appendix. B.3.

For the distance between the prior and posterior distributions of $Z$ and $\tau$, we can denote the loss as the following Kullback-Leibler divergence:

$$\mathcal{L}_{\mathrm{KL}}(\Phi; \boldsymbol{x}, \mathcal{R}) = \mathrm{KL}\left( q_\Phi(Z|\boldsymbol{x}) || p(Z) \right) + \mathrm{KL}\left( q_{\phi_\tau}(\tau|\mathcal{R}) || p(\tau) \right)$$

Finally, we train our model by minimizing the empirical loss on the training dataset $\mathcal{D}$ as follows:

$$\mathcal{L}(\Phi, \Theta; \mathcal{D}) = \frac{1}{|\mathcal{D}|} \sum_{i=1}^{|\mathcal{D}|} \left[ \mathcal{L}_{\mathrm{NLE}}(\Phi, \Theta; Y^{(i)}, \boldsymbol{x}^{(i)}, \mathcal{R}^{(i)}) + \mathcal{L}_{\mathrm{KL}}(\Phi; \boldsymbol{x}^{(i)}, \mathcal{R}^{(i)}) \right]. \quad (9)$$

## 4.3 GENERATION

To generate the segmentation results $\hat{Y}$, we can sample from the latent variables $Z$ and $\tau$ as follows:

$$p(\hat{\boldsymbol{y}}_{r_i}|\boldsymbol{x}, r_i) = p_{\theta_Y}(\hat{\boldsymbol{y}}_{r_i}|\tau_i, \boldsymbol{z}_i) q_{\phi_\tau}(\tau_i|r_i) q_{\phi_i}(\boldsymbol{z}_i|x). \quad (10)$$

For personalized segmentation results, we can specify the expert annotator $r \in \mathcal{R}$ to generate the corresponding segmentation results while the segmentation diversity is maintained by sampling from the latent representation $q_{\phi_i}(\boldsymbol{z}_i|x)$, which indicates the ambiguous boundaries of medical images.

For more diverse segmentation results, we can sample from the prior distribution of $\tau_*$, *i.e.*, $p(\tau_*) = \mathrm{Dir}(\alpha_*)$, to generate the segmentation results $\hat{\boldsymbol{y}}$, which indicates the subjective variations among expert annotators as follows:

$$p(\hat{\boldsymbol{y}}_*|\boldsymbol{x}) = p_{\theta_Y}(\hat{\boldsymbol{y}}_*|\tau_*, \boldsymbol{z}_i) p(\tau_*) q_{\phi_i}(\boldsymbol{z}_i|\boldsymbol{x}), \quad (11)$$

where $i$ denotes the class of a sample $\tau_*$ and is identified with the classifier $p_{\theta_r}(\mathcal{R}|\tau_*)$. Note that, being different from the $\tau_i$ in equation 10 which is personalized by the expert $r_i$, the class of $\tau_*$ is uncertain, resulting in random expert annotators and thus more diverse segmentations.

## 5 EXPERIMENTS

### 5.1 SETUP

**Dataset.** We evaluate our ProSeg on two medical image segmentation datasets: the nasopharyngeal carcinoma (**NPC**) (Wu et al., 2024) dataset and the lung nodule dataset (**LIDC-IDRI**) (Armato III et al., 2011). **LIDC-IDRI** dataset (Armato III et al., 2011) contains 1609 Computed Tomography (CT) images of 214 subjects with lung nodules, each image is provided with four expert annotations. Although twelve expert annotators are involved in the annotation, we rank the four given annotations by their segmentation area and assign them to four virtual annotators to simulate the consistent preference of the annotators following previous works (Wu et al., 2024; Zhang et al., 2020). Then the four virtual annotators are used as the expert annotators in the experiments and their preferencea are more consistent for all the images. Finally, following Wu et al. (2024), four-fold cross-validation is conducted on the dataset, where we split the data based on the patient ID to reduce bias in results arising from the similarity between adjacent lung nodule slices. **NPC** (Wu et al., 2024) is a more challenging dataset, where the expert annotators are not assigned according to the ranking but four real radiologists, *i.e.*, the preference of the annotators is more diverse and varies. NPC contains Magnetic Resonance Imaging (MRI) images of 120 subjects with nasopharyngeal carcinoma, where each image is annotated by four different expert annotators. Since one MRI image consists of multiple slices, we split the dataset according to subjects into 80 for training, 20 for validation and 20 for testing, which results in 5817 training slices, 1398 validation slices, and 1126 testing slices. More details for data preprocessing can be found in Appendix. C.1.1

**Evaluation metrics.** We employ four metrics to evaluate the performance of our ProSeg, including **Generalized Energy Distance (GED)** and **soft Dice score** $D_{soft}$ for diversity, as well as **match Dice score** $D_{match}$ and **maximum Dice score** $D_{max}$ for personalization. **GED** (Bellemare et al., 2017; Kohl et al., 2018) measures the diversity of the generated segmentation, which contains three parts: the difference between generated segments $d_{pp}$, the difference between generation and ground truth $d_{pa}$, and the difference between ground truths $d_{aa}$, *i.e.*, $GED = 2d_{pa} - d_{pp} - d_{aa}$. The lower GED score indicates the higher diversity of the generated segmentation. **Soft Dice score** $D_{soft}$ (Wang et al., 2023; Ji et al., 2021) measures the consistency between the annotations and generated segmentation, which is the mean Dice score between the average annotations and average predictions over a set of thresholds. **Maximum Dice score** $D_{max}$ measures the maximum overlap between the predicted and ground truth segmentation. In contrast, **Match Dice score** $D_{match}$ calculates the overlap between the predicted and ground truth segmentation with a constraint of one-to-one match (Wu et al., 2024). More details can be found in Appendix. C.1.2.

**Compared approaches.** We mainly compare our ProSeg with generation methods (Probabilistic U-Net (Kohl et al., 2018), D-persona (stage I) (Wu et al., 2024)) and personalized methods (CM-Global (Tanno et al., 2019), CM-Pixel (Zhang et al., 2020), TAB (Liao et al., 2023), Pionono (Schmidt et al., 2023), and D-persona (stage II) (Wu et al., 2024)). Besides, we also provide the results of U-Net trained on each expert annotator's annotations as the baseline.

### 5.2 EXPERIMENTAL RESULTS

First, we demonstrate the effectiveness of our ProSeg on the LIDC-IDRI dataset for both diversified and personalized medical image segmentation. Then, we conduct experiments on the more challenging NPC dataset, where the preference of the expert annotators varies more, to demonstrate the effectiveness of our ProSeg in a more practical setting.

#### 5.2.1 EVALUATION ON LIDC-IDRI

Table. 1 reports the performance of the different methods in terms of diversity, personalization, and personalized segmentation. Compared to generative methods, our ProSeg consistently achieves the best diversity performance with the lowest GED (0.1152) and the highest soft Dice score $D_{soft}$ (91.53%), which demonstrates its ability to generate a wide range of meaningful segmentation variations that can significantly improve diversity while maintaining high-quality segmentation. In terms of personalization performance, the $D_{max}$ of ProSeg reaches 91.03%, as it is able to effectively capture expert-specific segmentation patterns in the latent space. In terms of personalized segmentation

Table 1: Diversity and personalization on LIDC-IDRI dataset of individually trained U-Nets (Top), generation-based models (Middle), and personalized segmentation models (Bottom). The best results are highlighted in bold. #50 denotes the number of samples. I and II denote the stage.

| Method | Diversity Performance | | Personalization (%) | | Personalized Segmentation Performance (%) | | | | |
|---|---|---|---|---|---|---|---|---|---|
| | $GED \downarrow$ | $D_{soft} \uparrow (\%)$ | $D_{max} \uparrow$ | $D_{match} \uparrow$ | $D_{A1} \uparrow$ | $D_{A2} \uparrow$ | $D_{A3} \uparrow$ | $D_{A4} \uparrow$ | $D_{mean} \uparrow$ |
| U-Net ($A_1$) | 0.3062 | 86.59 | | | **87.80** | 87.47 | 85.49 | 80.67 | 85.36 |
| U-Net ($A_2$) | 0.2459 | 88.43 | N/A | | 87.16 | **89.08** | 88.59 | 85.15 | 87.50 |
| U-Net ($A_3$) | 0.2436 | 88.20 | | | 85.29 | 88.48 | **89.40** | 87.20 | 87.59 |
| U-Net ($A_4$) | 0.2962 | 85.83 | | | 80.80 | 85.48 | 88.22 | **88.90** | 85.85 |
| Prob. U-Net (#50) | 0.2168 | 88.80 | 88.87 | 88.81 | | | | | |
| D-Persona (I, #50) | 0.1358 | 90.45 | 91.37 | 91.33 | | | N/A | | |
| ProSeg (prior #50) | **0.1077** | **91.62** | **91.46** | **91.43** | | | | | |
| CM-Global | 0.2432 | 88.53 | 87.51 | 87.51 | 86.13 | 88.76 | 88.99 | 86.18 | 87.51 |
| CM-Pixel | 0.2407 | 88.64 | 87.72 | 87.72 | 85.99 | 88.81 | 89.31 | 86.77 | 87.72 |
| TAB | 0.2322 | 86.35 | 87.11 | 86.08 | 85.00 | 86.35 | 86.77 | 85.77 | 85.97 |
| Pionono | 0.1502 | 90.00 | 90.10 | 88.97 | 87.94 | 89.11 | 89.55 | 88.76 | 88.84 |
| D-Persona (II) | 0.1444 | 90.31 | 90.38 | 89.17 | 88.54 | 89.50 | 90.03 | 88.60 | 89.17 |
| ProSeg | **0.1152** | **91.53** | **91.03** | **90.25** | **89.49** | **90.27** | **91.01** | **90.23** | **90.25** |

accuracy, ProSeg achieves state-of-the-art (SOTA) performance with the highest average dice similarity ($D_{mean}$ = 90.25%) and consistently strong performance across annotators ($D_{A1}$ = 89.49%, $D_{A2}$ = 90.27%, $D_{A3}$ = 91.01%, $D_{A4}$ = 90.23%). Besides, by sampling the prior distribution of $q(\tau)$, our ProSeg (prior) achieves a better GED score, since more expert preferences are sampled. In summary, ProSeg achieves a better balance between diversity and personalization, ensuring that the generated segments are both aligned with experts and diverse.

### 5.2.2 EVALUATION ON NPC

We evaluated ProSeg's performance on the NPC dataset, where expert annotators' preferences vary more and segmentation is more challenging. As shown in Table. 3, ProSeg achieves superior performance in both diversity and personalization compared to existing methods. Specifically, as the diversity performance shown in columns 2-3, ProSeg achieves the lowest $GED$ and highest $D_{soft}$ scores among personalization methods, with an average improvement of 20.71% and 1.79% over previous SOTA methods, respectively. For personalization performance in columns 4-5, ProSeg achieves the highest $D_{max}$ (83.24%) and $D_{match}$ (82.13%) scores, outperforming all other baselines.

Although the performance of ProSeg on $GED$ scores is similar to that of previous SOTA generation methods, ProSeg is a better model that generates reliable and diverse segmentations for practical use, as we give the following explanations. First, ProSeg achieves the highest $D_{soft}$ score, indicating that ProSeg generates segmentations that are consistent with the average annotations. Then, to better understand the $GED$ score, we further analyze the $d_{pp}$, $d_{pa}$, and $d_{aa}$ scores of ProSeg and D-Persona. We find that ProSeg outperforms D-Persona on $d_{pa}$ (0.3482 *V.S.* 0.3648), which indicates that the segments generated by ProSeg match the ground truth better than D-Persona. Although ProSeg performs worse than D-Persona on $d_{pp}$ (0.2212 *V.S.* 0.1830), which indicates that ProSeg generates less diverse segmentations than D-Persona. The better performance of ProSeg on $d_{pa}$ and $D_{soft}$ indicates that ProSeg generates reliable segmentations that closely match the ground truth. More visual results can be found at Appendix. D.3.

### 5.3 ABLATION STUDY

We perform an ablation study to analyze the contributions of the two latent space components in ProSeg. Both the latent variables $\tau$ and $Z$ are crucial for achieving high performance in multi-rater medical image segmentation tasks as shown in Table. 2. Without the latent space $\tau$, the performance of ProSeg drops since the

| Method | | Diversity | | Personalization (%) | |
|---|---|---|---|---|---|
| $\tau$ | $Z$ | $GED \downarrow$ | $D_{soft} \uparrow (\%)$ | $D_{max} \uparrow$ | $D_{match} \uparrow$ |
| | | 0.3639 | 80.58 | 79.77 | 79.69 |
| ✓ | | 0.3091 | 82.10 | 81.19 | 80.98 |
| | ✓ | 0.2566 | 83.33 | 81.92 | 81.51 |
| ✓ | ✓ | **0.2272** | **84.24** | **82.36** | **82.07** |

Table 2: Ablation study on two latent spaces.

subjective variations among expert annotators are not modeled. Similarly, without the latent space $Z$, the performance of ProSeg also drops since the ambiguity in medical scans is not captured.

Table 3: Diversity and personalization performance on NPC dataset of individually trained U-Nets (Top), generation-based models (Middle), and personalized segmentation models (Bottom). The best results are highlighted in bold. #50 denotes the number of samples. I and II denote the stage.

| Method | Diversity Performance | | Personalization (%) | | Personalized Segmentation Performance (%) | | | | |
|---|---|---|---|---|---|---|---|---|---|
| | $GED \downarrow$ | $D_{soft} \uparrow (\%)$ | $D_{max} \uparrow$ | $D_{match} \uparrow$ | $D_{A1} \uparrow$ | $D_{A2} \uparrow$ | $D_{A3} \uparrow$ | $D_{A4} \uparrow$ | $D_{mean} \uparrow$ |
| U-Net ($A_1$) | 0.4531 | 76.48 | | | **85.93** | 72.09 | 72.11 | 75.79 | 76.48 |
| U-Net ($A_2$) | 0.4636 | 77.10 | | | 79.51 | **77.10** | 74.10 | 74.13 | 76.21 |
| U-Net ($A_3$) | 0.5057 | 75.40 | N/A | | 76.13 | 75.91 | **77.38** | 74.47 | 75.97 |
| U-Net ($A_4$) | 0.5606 | 71.17 | | | 78.79 | 71.05 | 70.80 | **74.33** | 73.74 |
| Prob. U-Net (#50) | 0.3614 | 80.94 | 82.42 | 79.95 | | | | | |
| D-Persona (I, #50) | **0.2133** | 83.24 | 81.52 | 80.25 | | | N/A | | |
| ProSeg (prior #50) | 0.2182 | **84.36** | 83.28 | 82.43 | | | | | |
| CM-Global | 0.3755 | 80.69 | 79.62 | 79.62 | 85.53 | 77.34 | 77.04 | 79.58 | 79.62 |
| CM-Pixel | 0.3678 | 80.92 | 79.86 | 79.86 | 85.09 | 77.08 | 77.37 | 79.91 | 79.86 |
| TAB | 0.3159 | 81.84 | 80.88 | 80.64 | 85.63 | 77.76 | 78.80 | 80.36 | 80.64 |
| Pionono | 0.3309 | 81.65 | 80.59 | 80.42 | 85.44 | 77.84 | 77.62 | 80.77 | 80.42 |
| D-Persona (II) | 0.2866 | 82.45 | 81.37 | 80.96 | 85.92 | 78.23 | 79.45 | 80.23 | 80.96 |
| ProSeg | **0.2272** | **84.24** | 82.36 | 82.07 | **87.48** | **79.39** | **80.66** | 80.77 | 82.07 |

## 6 DISCUSSION

**Why does ProSeg perform better? Compared to Generation-Based Methods.** Existing generation-based segmentation methods aim to generate diverse segmentation results by learning a latent space. However, they fail to capture expert-specific characteristics, leading to uncontrolled diversity that does not align with individual annotators. In contrast, ProSeg explicitly models expert variations using the latent variable $\tau$, allowing it to generate diverse yet expert-specific segmentation results. **Compared to Personalization Methods.** Personalization methods can produce expert-specific segmentation results but lack diversity. These methods typically overfit individual annotations, making it difficult to capture the inherent uncertainty in medical images. ProSeg overcomes this limitation by introducing the latent variable $Z$, ensuring that segmentation results are both expert-aligned and diverse. **Compared to Two-Stage Methods.** Recent two-stage methods attempt to balance diversity and personalization by separating these objectives into two distinct phases. However, this approach lacks a cohesive probabilistic framework, leading to potential inconsistencies. ProSeg unifies both objectives within a single probabilistic model, achieving better synergy between diversity and personalization.

**Performance gap between two datasets** The performance gap is primarily caused by the variation in the personal preferences of the experts. The variation within each expert is greater on the NPC dataset than on the LIDC dataset, as evidenced by dataset statistics, which makes it hard to train a U-Net. The ProSeg improvement relative to U-Net trained individually is more pronounced on the NPC dataset than on the LIDC-IDRI dataset (3.39% *V.S.* 1.45%). This indicates that ProSeg is more effective in capturing expert-specific characteristics on the NPC dataset, where the personal preferences of the experts are more diverse.

## 7 CONCLUSION

To tackle the problem of multi-rater medical image segmentation, we propose a probabilistic modeling framework, ProSeg, that generates both diverse and personalized segmentation results. ProSeg models the joint distribution of experts, annotations, and medical images using two latent variables, $\tau$ and $Z$, to capture expert-specific characteristics and image ambiguity. Our experiments on the LIDC-IDRI and NPC datasets demonstrate that ProSeg outperforms existing methods in terms of diversity and personalization, achieving state-of-the-art performance in multi-rater medical image segmentation. Our ablation study further confirms the importance of the latent variables $\tau$ and $Z$ in achieving high performance. Applying ProSeg to other medical image segmentation tasks and exploring the potential of our probabilistic modeling framework in other domains are promising directions for future research. For the limitation, although this work can be extended to other tasks beyond medical image segmentation, this paper only focuses on medical images.

## REPRODUCIBILITY STATEMENT

We have made extensive efforts to ensure the reproducibility of our work. Detailed dataset descriptions are provided in App. C.1.1, training configurations and hyperparameters are reported in App. C.2, and method details in App. B. Upon acceptance, we will release our models, together withtraining and inference code, to facilitate replication and further research.

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

# A APPENDIX

# B METHOD DETAILS

## B.1 NOTATIONS

Table 4: Summary of mathematical notions and corresponding notations.

| Notation | Notion |
|---|---|
| Lowercase letter | Scalar (e.g., a) |
| Bold lowercase letter | Vector (e.g., $\mathbf{a}$) |
| Capital letter | Matrix (e.g., $A$) |
| $\boldsymbol{x}, \hat{\boldsymbol{x}} \in \mathbb{R}^{h \times w}$ | Image, Generated Image |
| $\boldsymbol{y}_r, \hat{\boldsymbol{y}}_r \in \mathbb{R}^{d \times K}$ | Expert-specific ground-truth annotation; predicted annotation |
| $Y, \hat{Y} \in \mathbb{R}^{d \times K \times N}$ | Multi-rater ensemble annotations; Generated annotations |
| $\tau \in \mathbb{R}^{B \times N \times 1 \times 1}$ | Latent variable for modeling subjective variants among experts |
| $\boldsymbol{z} \in \mathbb{R}^{B \times C \times d}$ | Latent variable for ambiguity of medical images |
| $\mathcal{L}(\cdot); \mathcal{L}_{\mathrm{NLE}}(\cdot); \mathcal{L}_{\mathrm{KL}}(\cdot)$ | Expected loss; negative log-likelihood loss; KL loss |
| $p(\cdot\|\cdot), p_\theta(\cdot\|\cdot)$ | Posterior distribution, parameterized by $\theta$ |
| $q(\cdot\|\cdot), q_\phi(\cdot\|\cdot)$ | Variational distribution, parameterized by $\phi$ |
| $p(\boldsymbol{x}\|z_i); q(z_i\|\boldsymbol{x})$ | Distributions *w.r.t.* Images decoders;encoders |
| $p(\mathcal{R}\|\tau); q(\tau\|\mathcal{R})$ | Distributions *w.r.t.* class embedding; classifier |
| $p(\boldsymbol{y}_{r_i}\|\tau_i, z_i)$ | Distributions *w.r.t.* segmentation predictor |
| $\mathcal{N}(0, I); \mathrm{Dir}(\alpha_0)$ | Gaussian Distribution; Dirichlet distribution parameterized by $\alpha_0$ |
| $\mathcal{D}; \|\mathcal{D}\|$ | Dataset, the size of a dataset |
| $\Phi; \Theta$ | The sets of DNN parameters |
| $h \times w \to d; r$ | Image size; expert variable |

We also provide a notation table for a more clear understanding of our methods as shown in Table. 4.

## B.2 DERIVATION OF EQ. 4

The Eq. 4 in the main text as follows is derived based on: (1) The chain rule of probability. (2) Conditional independence: $Y_i$ depends only on $Z_i$ and $\tau_i$. (3) Mixture model structure, where $\tau_i$ acts as a latent selection variable.

$$p(Y|\boldsymbol{x}, \mathcal{R}, \tau, Z) = \prod_{i=1}^{N} p(\boldsymbol{y}_{r_i}|\tau_i, \boldsymbol{z}_i).$$

1. The chain rule of probability: from the chain rule, we can write the joint conditional probability of $Y$ given $\tau$ and $Z$ as Eq. 12. Since each $\boldsymbol{y}_{r_i}$ is conditionally independent given $\boldsymbol{z}_i$ and $\tau_i$, we can factorize the joint distribution as Eq. 13.

$$p(Y|\tau, Z) = p(\boldsymbol{y}_{r_1}, \boldsymbol{y}_{r_2}, \ldots, \boldsymbol{y}_{r_N}|\tau, Z) \tag{12}$$

$$p(Y|\tau, Z) = \prod_{i=1}^{N} p(\boldsymbol{y}_{r_i}|Z, \tau) \tag{13}$$

2. Local Markov Assumption: To further simplify the factorization, we assume that each $\boldsymbol{y}_i$ only depends on its local variables $\boldsymbol{z}_i$ and $\tau_i$ as Eq. 14. This is a common assumption in mixture models, where each data point is generated from a single component of the mixture. Thus, applying this to the previous equation, we get Eq. 15.

$$p(\boldsymbol{y}_i|Z, \tau) = p(\boldsymbol{y}_i|\boldsymbol{z}_i, \tau_i) \tag{14}$$

$$p(Y|\tau, Z) = \prod_{i=1}^{N} p(\mathbf{y}_{r_i}|\mathbf{z}_i, \tau_i) \tag{15}$$

### B.3 NEGATIVE LOG-LIKELIHOOD

The negative log-likelihood (NLL) is defined as the sum of the reconstruction loss of the image, the classification of the expert annotators, and the segmentation loss as follows:

$$\mathcal{L}_{\mathrm{NLE}}(\Phi, \Theta; Y, \mathbf{x}, \mathcal{R}) = \mathcal{L}_{\mathrm{recon}} + \mathcal{L}_{\mathrm{class}} + \mathcal{L}_{\mathrm{seg}} = \mathbb{E}\left[d(\hat{\mathbf{x}}, \mathbf{x})\right] + \mathbb{E}\left[\mathcal{R} \log \hat{\mathcal{R}}\right] + \mathbb{E}\left[Y \log \hat{Y}\right], \tag{16}$$

where, $\Phi = \{\phi_1, \ldots, \phi_N, \phi_\tau\}$, $\Theta = \{\theta_1, \ldots, \theta_N, \theta_\tau, \theta_Y\}$, $\hat{\mathbf{x}}$, $\hat{\mathcal{R}}$, and $\hat{Y}$ are the prediction of images, expert annotators, and annotations, respectively, and the expectations are over the variational distributions $q_{\phi_i}(\mathbf{z}_i|\mathbf{x})$ and $q_{\phi_\tau}(\tau|\mathcal{R})$. In the reconstruction loss $\mathcal{L}_{\mathrm{recon}}$, $d(\cdot, \cdot)$ denotes the mean squared error between the predicted image and the ground truth image. The classification loss $\mathcal{L}_{\mathrm{class}}$ is the cross-entropy loss between the predicted expert annotators and the ground truth expert annotators. The segmentation loss $\mathcal{L}_{\mathrm{seg}}$ is the cross-entropy loss between the predicted annotations and the ground truth annotations.

Here we give a detailed explanation of the NLE loss in Eq. 8. The $p(Y, \mathbf{x}, \mathcal{R}|Z, \tau)$ is the joint distribution of the annotations, images, and expert annotators given the latent variables $Z$ and $\tau$. We can factorize the joint distribution as follows:

$$p(Y, \mathbf{x}, \mathcal{R}|Z, \tau) = p(Y|Z, \tau)p(\mathbf{x}|Z)p(\mathcal{R}|\tau) \tag{17}$$

$$= \prod_{i=1}^{N} p(\mathbf{y}_{r_i}|\tau_i, \mathbf{z}_i) \prod_{i=1}^{N} p(\mathbf{x}|\mathbf{z}_i)p(r_i|\tau_i) \tag{18}$$

Taking the logarithm and negating it, we have:

$$-\ln p(Y, \mathbf{x}, \mathcal{R}|Z, \tau) = -\sum_{i=1}^{N} \ln p(\mathbf{y}_{r_i}|\tau_i, \mathbf{z}_i) - \sum_{i=1}^{N} \ln p(\mathbf{x}|\mathbf{z}_i) - \sum_{i=1}^{N} \ln p(r_i|\tau_i). \tag{19}$$

where $p(\mathbf{x}|\mathbf{z}_i)$ corresponds to the reconstruction loss $\mathcal{L}_{\mathrm{recon}}$, which measures the error between the predicted image $\hat{x}$ and the real image x. $p(\mathbf{y}_{r_i}|\tau_i, \mathbf{z}_i)$ Corresponds to segmentation loss $\mathcal{L}_{\mathrm{seg}}$, which measures the error between the predicted segmentation $\hat{Y}$ and the real segmentation Y. $p(r_i|\tau_i)$ corresponds to the classification loss $\mathcal{L}_{\mathrm{class}}$, which measures the error between the predicted expert labeling $\hat{R}$ and the real expert labeling $\mathcal{R}$. Finally, we have the NLE loss as Eq. 8.

## C EXPERIMENT DETAILS

### C.1 TRAINING DETAILS

#### C.1.1 DATASET

The input images from both the NPC and LIDC-IDRI datasets are center-cropped to a fixed size of $128 \times 128$. For NPC, we apply normalization to achieve zero mean and unit variance. To enhance training diversity, random flips, rotations, and noise perturbations are introduced as augmentation techniques. For LIDC-IDRI, we adopt the preprocessing strategy proposed by Wang et al. (2023) and employ standard flip and rotation operations for data augmentation.

The distribution of the rank of the annotations in the NPC datasets is shown in Fig. 4. The rank distribution of the NPC dataset is more diverse than that of the LIDC-IDRI dataset, which indicates that the NPC dataset is more challenging due to the diverse preferences of expert annotators, especially in the test dataset. For the LIDC-IDRI dataset, the rank distribution is consistent since we assign four virtual annotators according to the annotation area rank.

We also calculated the similarity of the annotations in the NPC test dataset between the expert annotators as shown in Table. 5. The distance is calculated by averaging the (1-IoU) over all slices. The distance matrix shows that the expert annotators have diverse preferences, which is consistent with the rank distribution in Fig. 4.

Table 5: Distance matrix of expert annotations in the NPC test dataset.

|          | Expert 1 | Expert 2 | Expert 3 | Expert 4 |
|----------|----------|----------|----------|----------|
| Expert 1 | 0.0000   | 0.6686   | 0.6696   | 0.6388   |
| Expert 2 | 0.6686   | 0.0000   | 0.6449   | 0.6650   |
| Expert 3 | 0.6696   | 0.6449   | 0.0000   | 0.6663   |
| Expert 4 | 0.6388   | 0.6650   | 0.6663   | 0.0000   |

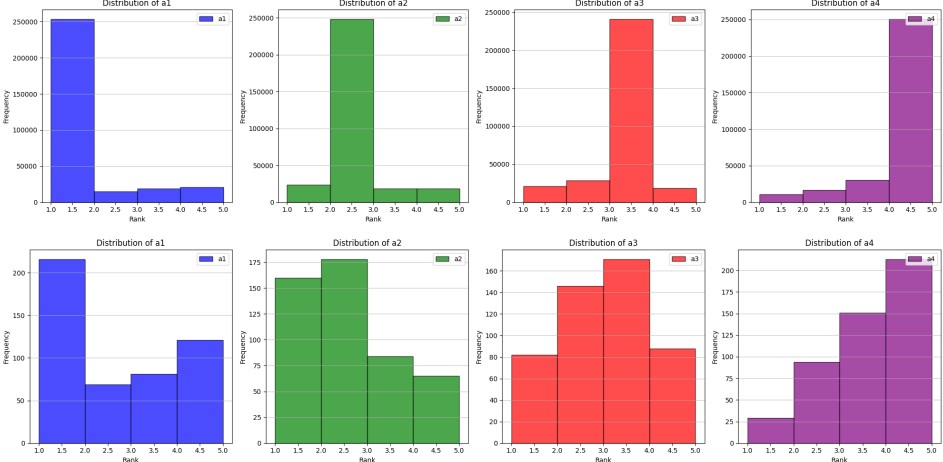

Figure 4: Expert annotator rank distribution of test (second row) and train (first row) datasets, where rank is obtained according to their annotation area.

### C.1.2 METRICS

**Experimental Metrics**    The four metrics used in the experiments are defined as follows:

- Generalized Energy Distance ($GED$) Bellemare et al. (2017); Kohl et al. (2018) as Eq. 20: $GED$ is used to measure the prediction diversity. A lower $GED$ indicates greater dispersion and variability in the segmentation results.

$$GED = \frac{2}{|Y||\hat{Y}|} \sum_{\boldsymbol{y} \in Y} \sum_{\hat{\boldsymbol{y}} \in \hat{Y}} [d(\boldsymbol{y}, \hat{\boldsymbol{y}})] - \frac{1}{|\hat{Y}||\hat{Y}|} \sum_{\hat{\boldsymbol{y}} \in \hat{Y}} \sum_{\hat{\boldsymbol{y}}' \in \hat{Y}} [d(\hat{\boldsymbol{y}}, \hat{\boldsymbol{y}}')] - \frac{1}{|Y||Y|} \sum_{\boldsymbol{y} \in Y} \sum_{\boldsymbol{y}' \in Y} [d(\boldsymbol{y}, \boldsymbol{y}')]$$

$$(20)$$

  where $Y$ and $\hat{Y}$ are the annotations and predicted segments, respectively. $d$ denotes the distance function $d(a, b) = 1 - IoU(a, b)$. $|\dot{|}$ indicates the number of elements in the set.

- Soft Dice Score ($D_{soft}$) Wang et al. (2023); Ji et al. (2021): $D_{soft}$ is used to evaluate the differences between the soft predictions and soft annotations. It is calculated by averaging the Dice scores over multiple thresholds as Eq. 21:

$$D_{soft} = \frac{1}{|\Gamma|} \sum_{\gamma \in \Gamma} Dice([\boldsymbol{y}_{soft} > \gamma_i], [\hat{\boldsymbol{y}}_{soft} > \gamma_i]),$$

$$(21)$$

  where $\gamma$ is a threshold selected from the set $\{0.1, 0.3, 0.5, 0.7, 0.9\}$ with $T = 5$, $\boldsymbol{y}_{soft} = \frac{1}{|Y_i|} \sum_{\boldsymbol{y} \in Y_i} \boldsymbol{y}$, and $\hat{\boldsymbol{y}}_{soft} = \frac{1}{|\hat{Y}_i|} \sum_{\hat{\boldsymbol{y}} \in \hat{Y}_i} \hat{\boldsymbol{y}}$

- Following Wu et al. (2024), Dice Max ($D_{max}$) and Dice Match ($D_{match}$): $D_{max}$ quantifies the optimal overlap between $Y$ and $\hat{Y}$, while $D_{match}$ further takes the one-to-one match from prediction to the expert annotator into account. Here, we give an example from Wu et al. (2024) to explain the two metrics. As shown in Fig. 5, $D_{max}$ averages the maximum scores of individual columns, while $D_{match}$ further constrains a one-to-one matching between the prediction and annotation sets. Here, $D_{max} = \{\mathbf{0.832}, \mathbf{0.842}, \mathbf{0.861}, \mathbf{0.863}\}$ and $D_{match} = \{\mathbf{0.832}, \mathbf{0.842}, \underline{0.841}, \mathbf{0.863}\}$.

|  | Annotations | | | |
|---|---|---|---|---|
|  | $r_1$ | $r_2$ | $r_3$ | $r_4$ |
| Predictions | 0.811 | 0.801 | 0.841 | 0.831 |
|  | 0.821 | 0.823 | 0.801 | 0.811 |
|  | **0.832** | 0.836 | **0.861** | 0.841 |
|  | 0.826 | 0.822 | 0.819 | **0.863** |
|  | 0.828 | **0.842** | 0.812 | 0.810 |

Figure 5: Example of $D_{max}$ and $D_{match}$ calculation in a given $4 \times 5$ Dice matrix.

**Random distance to two experts** We define the distance between experts as the IoU distance as follows:

$$d_{IoU}(S_1, S_2) = 1 - \frac{|S_1 \cap S_2|}{|S_1 \cup S_2|} \tag{22}$$

where $|S_1 \cap S_2|$ is the area of the intersection of $S_1$ and $S_2$, and $|S_1 \cup S_2|$ is the area of the union of $S_1$ and $S_2$. In Figure 1, we calculated the distance to two random experts for each slice in the NPC test dataset. The distance is calculated by averaging the (1-IoU) over all slices. For personalization methods, we take the four predictions from each expert. For the generation methods, we randomly take four experts as the determined experts. In one generation, we randomly select two experts and calculate the distance to them.

## C.2 MODEL DETAILS

We implement our ProSeg using PyTorch and all our experiments are conducted on a computing cluster with 8 GPUs of NVIDIA GeForce RTX 4090 24GB and CPUs of AMD EPYC 7763 64-Core of 3.52GHz. All the inferences are conducted on a single GPU of NVIDIA GeForce RTX 4090 24GB. The image encoder and decoder are implemented with a U-Net architecture, while the class embedding and classifier are implemented with a fully connected neural network. The segmentation predictor is implemented with a convolutional neural network. The model is trained with the Adam optimizer with a learning rate of $1e-4$ and a batch size of 12. The model is trained for 100 epochs with early stopping based on the validation loss. The latent space $Z$ and $\tau$ are set to $128 * 128$ and 8, respectively. The concentration parameter $\alpha$ is set to 1.0. The model is trained with the negative log-likelihood loss and the Kullback-Leibler divergence loss. The model is evaluated on the test dataset with the metrics described in the previous section.

## C.3 RESOURCE REQUIREMENTS

We have included a detailed comparison of computational complexity and training efficiency in the table below. As shown, **ProSeg** achieves a good balance in memory usage and inference/training time compared to other methods. This demonstrates its practicality for deployment in real-world scenarios.

| Method | Train Memory (MB) | Train Time (s) | Infer Memory (MB) | Infer Time (s) |
|---|---|---|---|---|
| CM-Global | 402.06 | 10.39 | 201.41 | 0.23 |
| CM-Pixel | 404.16 | 15.52 | 201.74 | 0.26 |
| Pionono | 410.71 | 25.11 | 202.42 | 1.07 |
| D-Persona (I) | 471.22 | 22.75 | 916.54 | 2.57 |
| D-Persona (II) | 407.29 | 19.95 | 241.91 | 0.99 |
| **ProSeg** | 411.32 | 17.51 | 202.42 | 1.05 |

Table 6: Comparison of training and inference efficiency across different methods.

# D ADDITIONAL RESULTS

We provide more quantitative and visual results here for a better explanation of the performance comparison.

Table 7: Performance of U-Net models on the NPC test dataset.

| Model | $D_{A1}\uparrow$ | $D_{A2}\uparrow$ | $D_{A3}\uparrow$ | $D_{A4}\uparrow$ | $D_{mean}\uparrow$ |
|---|---|---|---|---|---|
| U-Net ($A_1$) | **85.93** | 72.09 | 72.11 | 75.79 | 76.48 |
| U-Net ($A_2$) | 79.51 | **77.10** | 74.10 | 74.13 | 76.21 |
| U-Net ($A_3$) | 76.13 | 75.91 | **77.38** | 74.47 | 75.97 |
| U-Net ($A_4$) | 78.79 | 71.05 | 70.80 | **74.33** | 73.74 |

Table 8: Distance of experts in the NPC test dataset.

| | $A_1$ | $A_2$ | $A_3$ | $A_4$ |
|---|---|---|---|---|
| $A_1$ | 0.0000 | 0.6686 | 0.6696 | 0.6388 |
| $A_2$ | 0.6686 | 0.0000 | 0.6449 | 0.6650 |
| $A_3$ | 0.6696 | 0.6449 | 0.0000 | 0.6663 |
| $A_4$ | 0.6388 | 0.6650 | 0.6663 | 0.0000 |

## D.1 RESULTS DETAILS

**Performance of U-Net trained on each expert annotator's annotations.** We provide the performance of U-Net trained on each expert annotator's annotations as the baseline. The results are shown in Table. 3. Here, we give an explanation for their performance variations. We have two findings by comparing their performance and distance as shown in Table. 8:

1. The distance between expert annotations is consistent with the learned U-Net (each column). When the distance between the annotations of two experts is closer, their performance is more similar on one test set. For the test set of expert $A_3$, $A_4$ is more different with $A_3$ than $A_2$ (0.6663($A_4$) *V.S.* 0.6449($A_2$)). Therefore, the U-Net trained on $A_2$ performs better than that trained on $A_4$ on the test set of $A_3$ (70.80($A_4$) *V.S.* 74.10($A_2$)).

2. Training a U-Net to segment small target is harder than big targets. As shown in Table. 4, the segmentation area of $A_4$ is smaller than the others, thus it performs much worse than the others.

Table 9: Decomposed $GED$, including $d_{pp}$, $d_{pa}$, and $d_{aa}$.

| Method | $GED\downarrow$ | $d_{pp}\uparrow$ | $d_{pa}\downarrow$ | $d_{aa}(Constant)$ |
|---|---|---|---|---|
| Prob. U-Net | 0.3614 | 0.0075 | 0.3320 | 0.2951 |
| D-Persona (I) | **0.2133** | **0.2212** | 0.3648 | 0.2951 |
| ProSeg (prior) | 0.2182 | 0.1865 | **0.3499** | 0.2951 |
| CM-Global | 0.3755 | 0.0000 | 0.3353 | 0.2951 |
| CM-Pixel | 0.3678 | 0.0000 | 0.3314 | 0.2951 |
| TAB | 0.3159 | 0.0578 | 0.3344 | 0.2951 |
| Pionono | 0.3309 | 0.0317 | **0.3289** | 0.2951 |
| D-Persona (II) | 0.2866 | 0.0913 | 0.3365 | 0.2951 |
| ProSeg | **0.2272** | **0.1739** | 0.3482 | 0.2951 |

## D.2 QUANTITATIVE RESULTS

**GED score comparison.** For a better understanding of the GED score of all methods, we provide the decomposed GED score of all methods in Table. 3. The GED score is decomposed into $d_{pp}$, $d_{pa}$, and $d_{aa}$, which indicates the diversity of the generated segmentations, the difference between the generated segmentations and the ground truth, and the difference between the ground truths,

respectively. As shown in Table 9. GED balances the diversity of the generated segmentations and the consistency with the ground truth. ProSeg achieves the highest $d_{pp}$ score among personalization methods, which indicates that ProSeg generates diverse segmentations. Although ProSeg performs worse than previous personalization methods on $d_{pa}$, there is only a small difference. Both the high $d_{pp}$ score and relatively low $d_{pa}$ scores indicate that ProSeg generates both diverse and reliable segmentations.

**statistical significance testing** Compared to D-Persona(II) on NPC dataset, ProSeg obtained better performance on all the metrics, especially significantly better on $GED$ (p=1e-6), $D_{soft}$ (p=1e-5), $D_{max}$ (p=0.026), $D_{match}$ (p=0.006), $D_{A1}$ (p=0.044), and $D_{mean}$ (p=0.006) with p < 0.05. On the LIDC-IDRI dataset, ProSeg also obtained better performance on all the metrics, especially significantly better on $GED$ (p=5e-6), $D_{soft}$ (p=0.036), $D_{match}$ (p=0.046), and $D_{A4}$ (p=0.021) with p < 0.05. Compared to the other methods, i.e., Pionono, TAB, CM-Pixel, and CM-Global, ProSeg achieved better performance on all the metrics (p < 0.05).

## D.3 VISUAL RESULTS

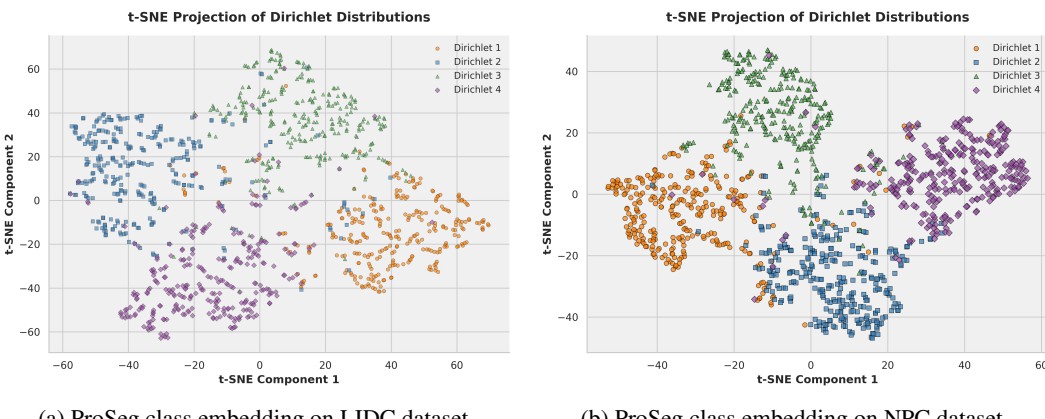

(a) ProSeg class embedding on LIDC dataset.  (b) ProSeg class embedding on NPC dataset.

Figure 6: Class embedding distribution

**Dirichlet distribution of Expert.** We randomly sample 300 samples from the posterior distribution of $p(\tau|\boldsymbol{r})$ for each expert. Then we use tsne (Van der Maaten & Hinton, 2008) to project the 8-dimension sample into 2-dimension for visualization as shown in Fig. 6. The class embedding of ProSeg trained on both datasets can be identified clearly, which indicates that the experts are different with respect to their preferences. Since some of the preferences of these experts are similar, some of the embeddings are mixed. By comparing Fig 6a with Fig. 6b. We have the following findings: **(1)** The width of distribution on the LIDC dataset is larger than that on the NPC dataset, which indicates that the ProSeg trained on the LIDC dataset can generate more diverse segmentation. **(2)** The diversity is also demonstrated in Table. 1 and Table. 3, where the GED score for ProSeg trained on LIDC dataset is 0.1152, while on NPC dataset is 0.2272. **(3)** the distribution of each class for NPC is more centralized. This is because the expert annotator in the NPC dataset is the real expert, while the expert annotators in the LIDC dataset are virtual experts, which is obtained by assigning 12 real experts to 4 virtual experts by their segmentation area ranking for each image. In addition, the four categories in the NPC dataset are relatively closer because their preferences overlap.

**Distance to two random experts.** For a clearer understanding of the distance between two random experts, we provide more visual results as shown in Fig. 7. The results are (a) the average distance to two random experts for each objective in the NPC test dataset of all methods (b) the distance to two random experts for each objective in the NPC test dataset of generation methods, and (c) the distance to two random experts for each objective in the NPC test dataset of personalization method. Fig. 7(a) shows that our ProSeg is best in diversity and personalization. Fig. 7(b) shows that the prior sampling of our ProSeg is better than other generation methods. Fig. 7(c) shows that ProSeg is better

than other personalization methods. All these results indicate that ProSeg is the closest method to the gold standard.

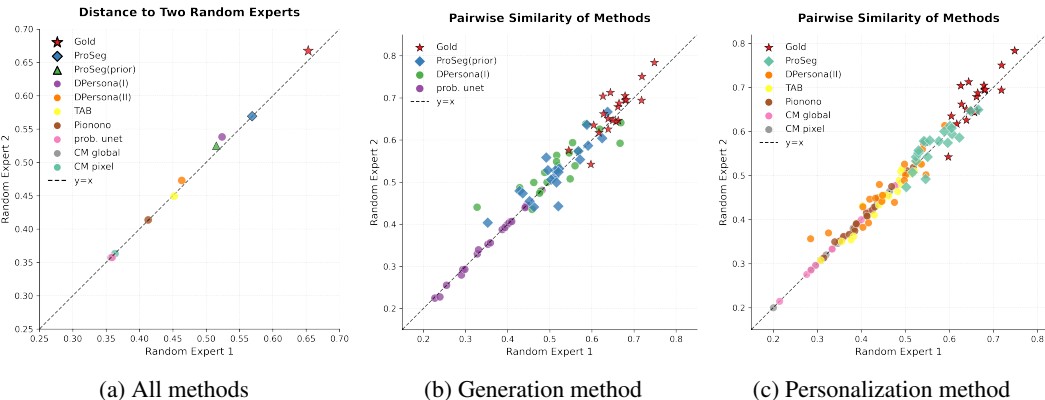

(a) All methods      (b) Generation method      (c) Personalization method

Figure 7: Performance on multi-rater medical image segmentation

**Distance between two experts.** For each pair of experts, we also calculate their distance. The distribution is shown in Fig. 8. The distance distribution of the Gold standard and our ProSeg are the most similar. By sampling the prior distribution of $p(\tau)$, *i.e.*, ProSeg (prior), the width of the distance distribution is greater than ProSeg. Compared with other methods, the distance distribution of our ProSeg is most similar to the Gold standard.

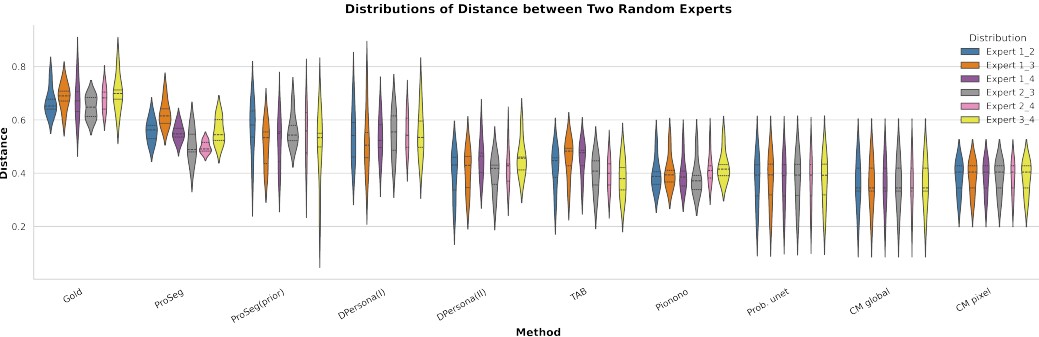

Figure 8: Distribution of pair distance between two experts.

**Visual comparison of segmentation results.** We provide visual comparisons of the segmentation results of all methods on the NPC dataset in Fig. 9 and Fig. 10, where different colors indicate the segmentation is obtained by different expert annotators. The segmentation results of ProSeg are more diverse and personalized than those of other methods. The segmentation results of ProSeg are more consistent with the ground truth while maintaining diversity among the generated segmentations. The results demonstrate that ProSeg effectively captures expert-specific characteristics and generates diverse segmentation results. For some methods, the segmentation from all the experts is the same, which means the diversity is poor. In Fig. 6b, the second row shows the segmentation from our ProSeg, the third row shows the segmentation from the DPersona (stage 1) and the fourth row shows the segmentation from the DPersona (stage 2). For the second image, in the gold standard, three experts give segmentation containing two separate parts. Our ProSeg captures the character, while other models can hardly capture this difference, and generation methods can not tell which expert gives the two-part segmentation as shown in the figure that the color of the two-part segmentation is different from the Gold standard.

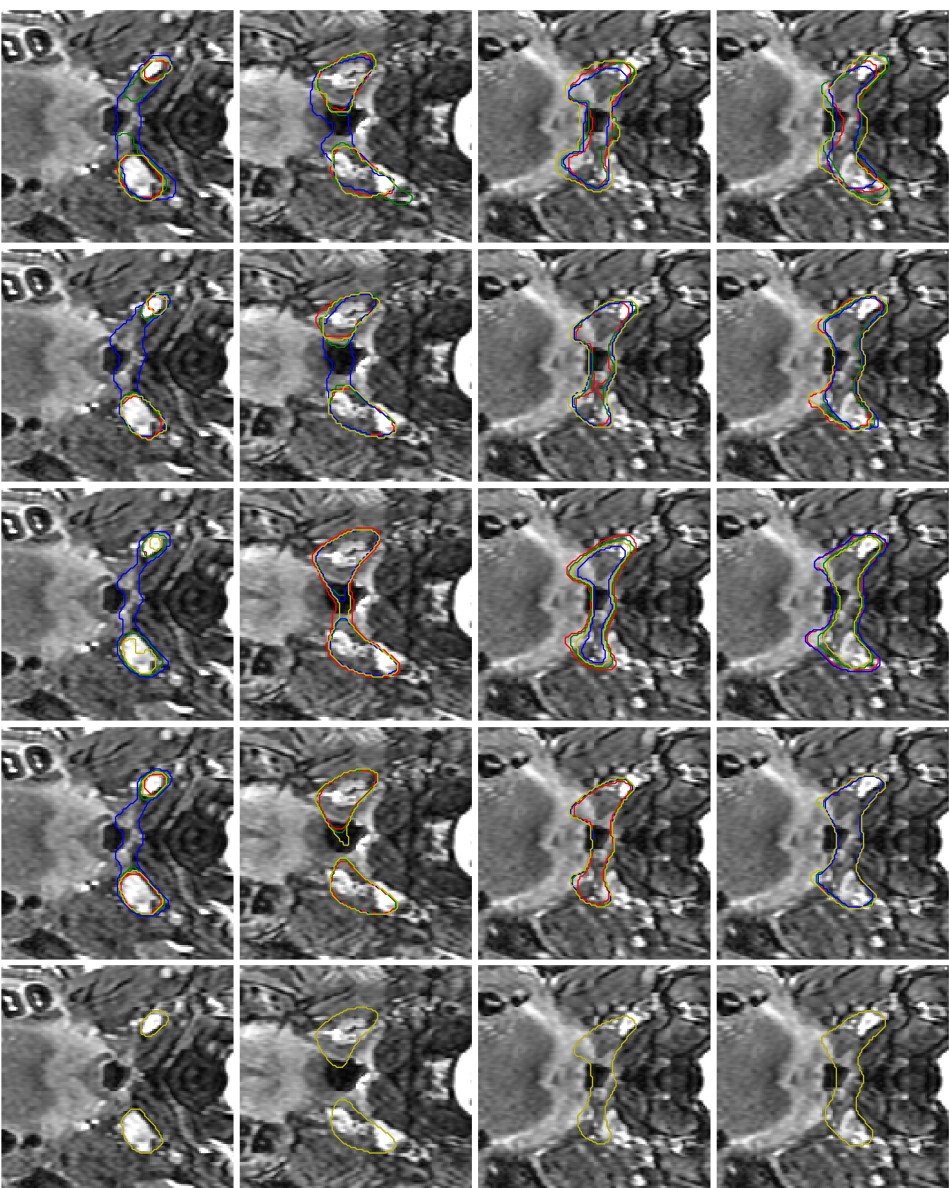

Figure 9: Visual results of segmentation on NPC dataset. Each row from the top to bottom indicates the Gold standard, ProSeg, DPersona (stage 1), DPersona (stage 2), and CM global.

## E  THE USE OF LARGE LANGUAGE MODELS (LLMS)

We use LLMs (GPT-5.0 and Gemini 2.5 pro) to polish our writing and check our grammar only.

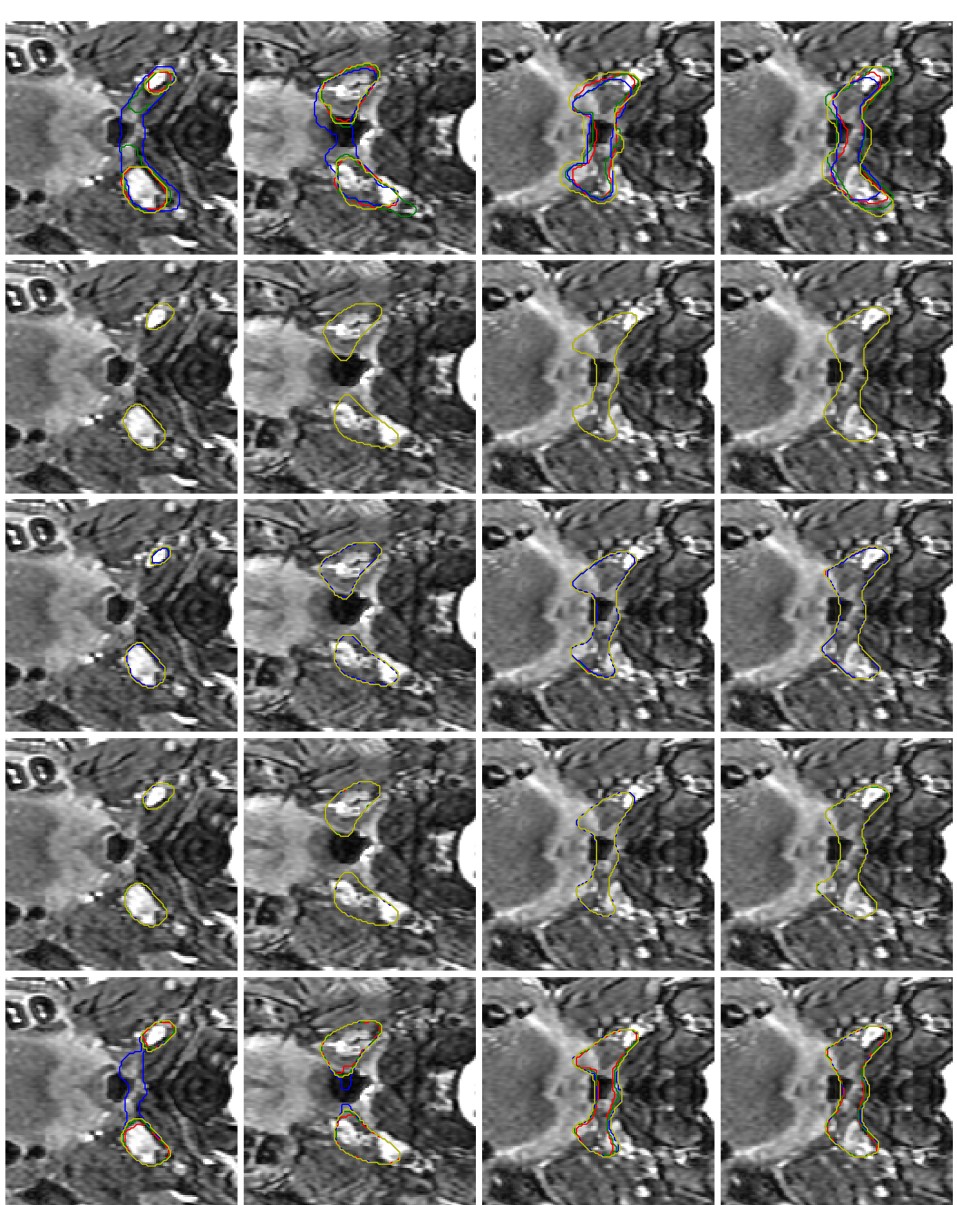

Figure 10: Visual results of segmentation on NPC dataset. Each row from the top to bottom indicates the Gold standard, CM pixel, Pionono, Probabilistic U-Net, and TAB.

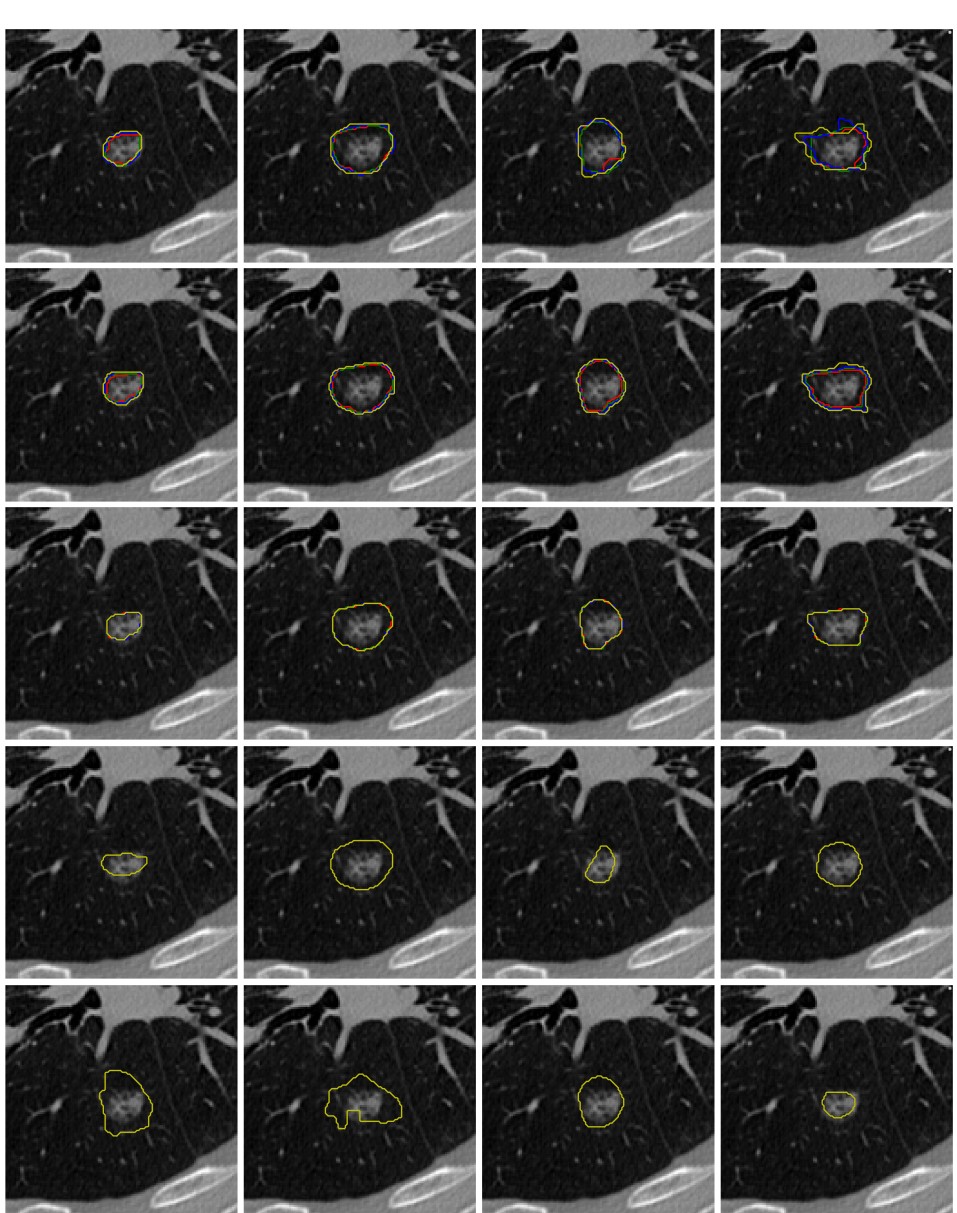

Figure 11: Visual results of segmentation on LIDC-IDRI dataset. Each row from the top to bottom indicates the Gold standard, ProSeg, ProSeg (prior), CM global, and CM pixel.

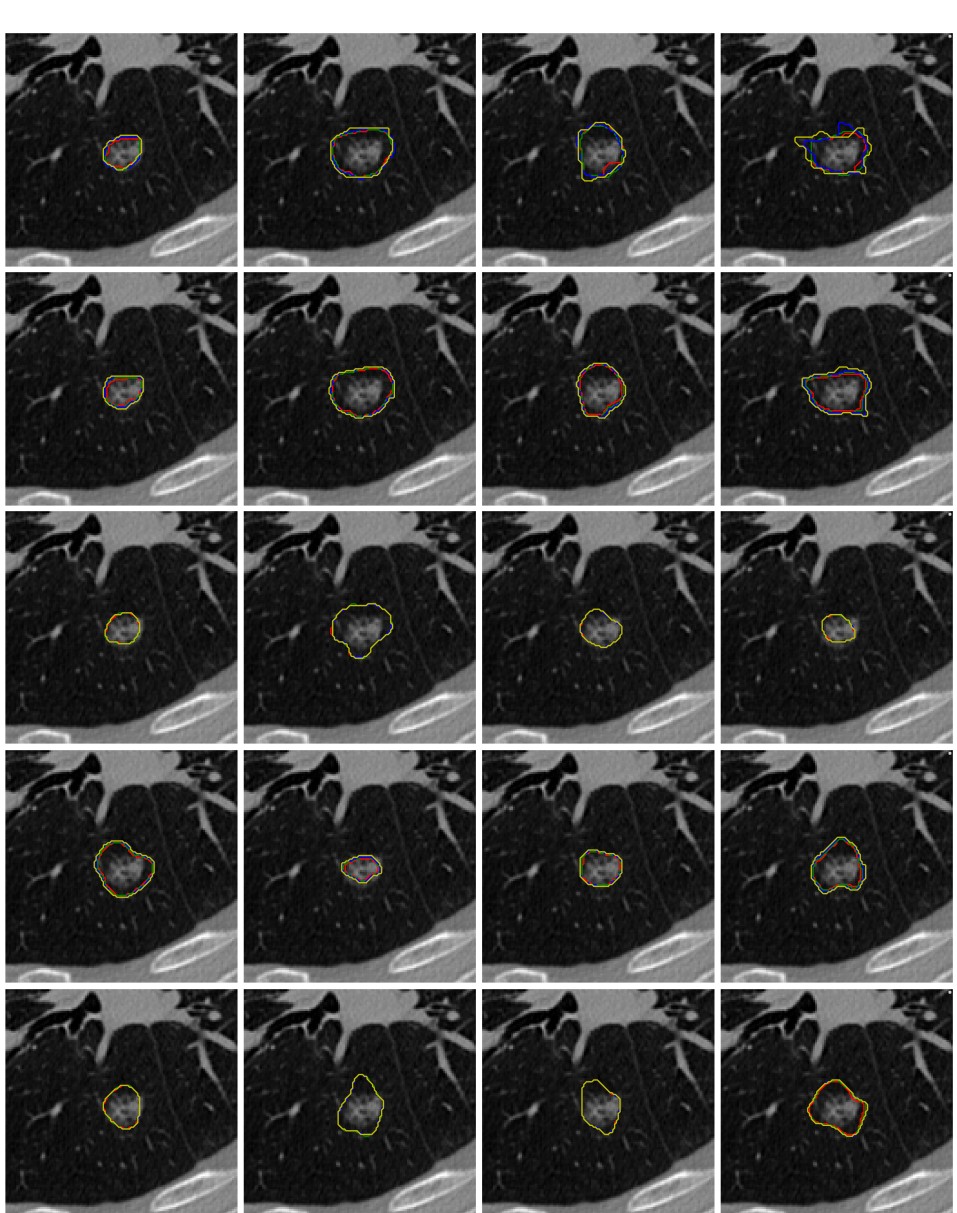

Figure 12: Visual results of segmentation on LIDC-IDRI dataset. Each row from the top to bottom indicates the Gold standard, ProSeg, Prob. U-Net, DPersona (stage 1), and DPersona (stage 2).

