# OpenReview forum: "Probabilistic Modeling of Multi-rater Medical Image Segmentation for Diversity and Personalization"
_ICLR.cc/2026/Conference — Submitted to ICLR 2026_

### Official Review · Reviewer_Cnm2 · 2025-10-19

**Soundness:** 4
**Presentation:** 3
**Contribution:** 4
**Rating:** 8
**Confidence:** 5

**Summary:**

This paper presents a probabilistic graphical model, ProSeg, for multi-rater medical image segmentation. The key design lies in introducing two latent variables to separately model expert-specific preferences and data ambiguity, allowing simultaneous generation of diverse and personalized segmentation results. The contribution is clear and well-motivated, and the method achieves consistently superior performance compared to related works on two public multi-rater datasets.

**Strengths:**

Clear and interesting motivation and contributions.

Better performance than other works.

An end-to-end framework to ensure efficiency.

Good paper written.

**Weaknesses:**

The task does not have real groundtruth to supervise the diverse segmentation outcomes. It remains unclear how the model guarantees prediction reliability and prevents hallucinated or implausible segmentations during sampling.

Given the rapid advances in autoregressive (AR) and diffusion-based generative segmentation models, it would be important to evaluate how the proposed probabilistic framework performs when integrated with or compared to these stronger generative backbones.

The current design involves two latent variables. It would be valuable to explore the scalability of the probabilistic model with additional latent factors. For example, incorporating random variables to intra-expert uncertainty.

In the current ProSeg formulation, each expert is assumed to be conditionally independent, modeled by its own latent variable. Also, how to model the relationships among different experts? Also, how about modeling this setting into a causal graph model and putting a shared latent condition?

The data ambiguity is modeled by a Gaussian distribution, and the expert priors are Dirichlet distributions. However, there is no clear motivation for adopting the two parameterized distributions. What is the advantage?

This work follows the typical setting, evaluation, datasets as in prior works. The unique contribution should be further highlighted in the main content.

The dataset should be cited with its correct reference. Wu et al., "Dataset, Challenge, and Evaluation for Tumor Segmentation Variability," ACM MM 2024.

**Questions:**

See the above weaknesses.

---

> ### Author Response · Authors · 2025-11-14
> **Response to Reviewer Cnm2: Appreciation and Discussion on Model Extensions and Priors**
>
> We sincerely thank the reviewer for the encouraging evaluation and for recognizing the novelty, motivation, and efficiency of our work. We appreciate the insightful suggestions regarding generative backbones and latent variable extensions.
>
> > W1: *"Unclear how the model guarantees reliability... prevents hallucination."*
>
> **R1**: This is a crucial point in generative segmentation. We address reliability through two mechanisms:
> -	Metric Validation: We rely on Soft Dice ($D_{soft}$) and $d_{pa}$ (distance to average annotation) alongside GED. As shown in Tables 1 and 3, ProSeg achieves state-of-the-art $D_{soft}$ scores (e.g., 84.24% on NPC), significantly higher than baselines. This indicates that while our generated samples are diverse (high $d_{pp}$), they remain highly consistent with the expert consensus, minimizing "hallucinations."
> -	Latent Regularization: The reconstruction loss $\mathcal{L}_{recon}$ (Eq. 8) acts as a regularizer, ensuring the latent space $Z$ preserves structural fidelity to the input image $X$.
>
> > W2: *Comparison/Integration with Autoregressive (AR) and Diffusion models.*
>
> **R2**: We appreciate this insightful suggestion. We would like to clarify two key points:
>
> - ProSeg is a Model-Agnostic Framework: ProSeg is fundamentally a Probabilistic Graphical Model (PGM) framework, which is orthogonal to the specific neural backbone. While we implemented it using a CNN (U-Net) in this work, the PGM formulation (modeling latent $\tau$ and $Z$) supports arbitrary backbones. It can be readily integrated with ViTs, Mamba, or even Diffusion/AR models as the feature extractor or generator, offering significant flexibility for future expansion.
>
> - Fair Comparison & Efficiency: We deliberately chose the standard U-Net backbone to ensure a fair comparison with established baselines (e.g., Probabilistic U-Net, D-Persona), thereby isolating the performance gains attributable to our probabilistic modeling rather than a more powerful backbone. Furthermore, compared to Diffusion models which require slow iterative sampling, our current implementation achieves SOTA diversity with real-time inference (1.05s), offering a superior clinical trade-off.
>
>
> > W3: *Scalability to additional latent factors (e.g., intra-expert uncertainty).*
>
> **R3**: We appreciate this insightful suggestion. Our PGM framework is indeed extensible. We can introduce a third latent variable, e.g., $u \sim \mathcal{N}(0, \sigma_r)$, conditioned on the expert $r$, to model the intra-rater variance (inconsistency of the same expert over time). We will discuss this promising direction as future work.
>
> > W4: *Conditional independence of experts / Shared latent condition.*
>
> **R4**: Our current formulation (Eq. 6) assumes experts are conditionally independent in decision-masking given the latent variables $\tau$ and $Z$. This follows the standard Mixture-of-Experts assumption to make the variational inference tractable. However, we agree that modeling causal relationships (e.g., Expert A influences Expert B) using a Directed Acyclic Graph (DAG) or a shared "hospital-level" latent condition is a fascinating extension for future hierarchical probabilistic models.
>
> > W5: *Motivation for Gaussian (Z) and Dirichlet ($\tau$) priors.*
>
> **R5**: We selected these priors based on the physical nature of the uncertainty sources:
> - Gaussian for $Z$: Standard choice in VAEs to model continuous spatial ambiguity (e.g., gradual blurring of tumor boundaries).
> - Dirichlet for $\tau$: The theoretically correct conjugate prior for categorical/multinomial distributions. It naturally models "mixtures" of expert preferences (e.g., a style lying on the simplex between "Conservative" and "Aggressive" prototypes), allowing us to sample valid intermediate styles.
>
> > W6: *Citation correction.*
>
> **R6**: We apologize for the oversight. We will correctly cite Wu et al., "Dataset, Challenge, and Evaluation for Tumor Segmentation Variability," ACM MM 2024 in the final version.

---

> > ### Comment · Reviewer_Cnm2 · 2025-11-26
> > **Thanks for the response**
> >
> > I thank the authors for their response. Overall, I appreciate the idea of employing a single-stage training framework alongside two parameterized distributions for modeling both data and annotators. Here are some suggestions.
> >
> > It is difficult to justify the assumption that experts are independent. The use of a Dirichlet distribution appears to allow only interpolation among existing annotators, while extrapolation to unseen experts remains challenging under the current formulation. This raises concerns regarding some of the claims made in the rebuttal.
> >
> > Moreover, examining different subsets of real datasets often yields varying similarity matrices, suggesting that experts naturally share overlaps and are not independent. Introducing a random perturbation or a mechanism to capture such dependencies may be necessary.
> >
> > Overall, I encourage the authors to explore these aspects further in future revisions, and I will maintain my original score.

---

### Official Review · Reviewer_wpoR · 2025-10-24

**Soundness:** 2
**Presentation:** 2
**Contribution:** 2
**Rating:** 2
**Confidence:** 4

**Summary:**

This paper proposes ProSeg, a probabilistic variational framework designed to model both inter-rater diversity and annotator-specific personalization in medical image segmentation. The model introduces two latent variables, τ and Z, to capture annotator preference and data ambiguity, respectively, and optimizes a unified evidence lower bound (ELBO).
Experiments are conducted on two datasets, LIDC-IDRI and NPC, where the method is compared with several baselines such as Probabilistic U-Net, TAB, and D-Persona.
The authors claim that ProSeg is the first unified probabilistic approach capable of simultaneously capturing diversity and personalization among multiple annotators.
While the topic is meaningful and the writing is generally clear, the paper suffers from fundamental conceptual and methodological flaws that seriously weaken its contributions and undermine the reliability of its results.

**Strengths:**

1. The paper addresses an important problem in uncertainty-aware medical image segmentation by attempting to unify the modeling of inter-rater diversity and annotator personalization within one probabilistic framework.

2. The probabilistic formulation and ELBO derivation are mathematically consistent, and the structure of the manuscript is clear and easy to follow.

3. The visualizations and tables are well presented, and the authors compare with several existing multi-rater methods.

**Weaknesses:**

1. The use of pseudo-annotators in the LIDC-IDRI experiments is not representative. Instead of using real annotator identities, the authors artificially construct four “virtual doctors” by sorting masks according to lesion area. This invalidates the claim that the method learns human-specific annotation styles. What the model effectively learns is merely area-based morphological variation, not true inter-rater differences. As a result, the reported personalization metrics are not meaningful, and the experimental foundation of the paper becomes questionable.

2. The theoretical formulation relies on several unrealistic independence assumptions such as p(Y∣X,R,τ,Z)≈p(Y∣τ,Z) and a factorized prior over latent variables. None of these assumptions are justified empirically, nor is there ablation analysis to confirm that they are harmless.

3. The variational family and prior selections are also rather arbitrary: the authors fix Z as a standard Gaussian and τ as a symmetric Dirichlet distribution with α₀ = 1, yet provide no heuristic or sensitivity analysis to justify these choices. It is unclear why the symmetric prior is reasonable for modeling annotator heterogeneity, how different α values would influence the learned distribution, or how the model behaves under varying numbers of raters or classes. Given that the paper explicitly aims to model individual annotator differences, such prior rigidity and the absence of sensitivity evaluation make the entire probabilistic formulation insufficiently validated.

4. The paper lacks any efficiency analysis—no training time, GPU memory, or parameters, inference time are reported. This omission makes it impossible to judge the practical scalability of the framework.

5. The experimental validation is narrow and does not demonstrate generalization. Only two relatively small datasets are used (LIDC-IDRI and NPC). There is no evaluation on other publicly available multi-rater datasets (e.g., RIGA[a], QUBIQ[b], and SUN-SEG[c]), no cross-dataset or cross-modality experiments, and no statistical significance reporting.

6. The presentation raise further concerns. The paper exhibits poor adherence to academic standards: the reference list contains numerous formatting inconsistencies, missing venues and publishers, and the authors do not provide a reproducibility statement or code prior to acceptance. These presentation and transparency issues further reduce the credibility of the work.

[a] Almazroa, Ahmed, et al. "Agreement among ophthalmologists in marking the optic disc and optic cup in fundus images." International ophthalmology 37.3 (2017): 701-717.

[b] Li, Hongwei Bran, et al. "Qubiq: Uncertainty quantification for biomedical image segmentation challenge." arXiv preprint arXiv:2405.18435 (2024).

[c] Ji, Ge-Peng, et al. "Video polyp segmentation: A deep learning perspective." Machine Intelligence Research 19.6 (2022): 531-549.

**Questions:**

See weaknesses.

---

> ### Author Response · Authors · 2025-11-14
> **Response to Reviewer wpoR: Correcting Critical Factual Oversights and Justifying Experimental Design**
>
> We thank the reviewer for acknowledging the meaningful topic and clear presentation. However, we must respectfully point out that the review contains **several critical factual oversights** regarding our manuscript's content, which appear to have fundamentally and negatively influenced the assessment.
>
> > W1: *"Pseudo-annotators... invalidates the claim."*
>
> **R1**: We respectfully point out that This concern is based on a **misunderstanding** of the standard experimental protocol in this specific sub-field.
> - **Standard Benchmark Protocol**: The use of "virtual experts" (ranking annotations by area) is the strictly followed standard protocol in the multi-rater personalization community (e.g., D-Persona [Wu et al., CVPR 2024], CM-Pixel [Zhang et al., NeurIPS 2020]). Adhering to this protocol is mandatory to ensure a fair quantitative comparison (e.g., $D_{match}$, $D_{max}$) against these SOTA baselines (as shown in Table 2). Abandoning this would make comparison with existing literature impossible.
> - **Validation on Real Radiologists**: Our NPC dataset results (Table 3) are based on **4 real radiologists** without any artificial regrouping or ranking. ProSeg achieves significant improvements here ($D_{mean}$ +1.3% over SOTA), empirically proving that our model effectively captures true, complex human inter-observer variability and is not limited to simple area-based heuristics.
>
> > W2 & W3: *Independence assumptions and Prior selections.*
>
> **R2 & R3**:
> 1. Independence: Our factorization (Eq. 6) follows the standard Variational Mixture-of-Experts formulation. We assume that given the latent expert style $\tau$ and image content $Z$, the **specific segmentation realization is conditionally independent of the raw expert ID. This is a necessary and standard relaxation to make the ELBO tractable while maximizing the joint likelihood of observing** consistent expert-image pairs.
> 2. Priors: We selected priors based on the physical nature of the factors:
> - **Feasibility**: Inappropriate priors can render variational inference intractable. By selecting Gaussian and Dirichlet priors, we can use variational distributions from the same families to approximate the posteriors, ensuring analytical tractability as recommended in standard variational inference frameworks.
> - **Gaussian for $Z$**: A standard choice in VAEs (e.g., Probabilistic U-Net) to model continuous spatial ambiguity (e.g., blurred boundaries). Beyond VAEs, diffusion models also embed images within a Gaussian latent space, further demonstrating that a Gaussian prior is a robust and effective choice for modeling the distributions of real-world images.
> - **Dirichlet for $\tau$**: The theoretically correct conjugate prior for categorical distributions (expert identity). It naturally models "mixtures" of expert preferences (e.g., a style that is 80% Expert A and 20% Expert B).
> - **Sensitivity**: We used a non-informative symmetric prior ($\alpha_0=1$) specifically to avoid introducing subjective bias, allowing the data to predominantly determine the learning of expert clusters.
>
> > W4: *"Lacks any efficiency analysis... impossible to judge scalability."*
>
> **R4**: This statement is **factually incorrect**. We have explicitly provided a detailed efficiency analysis in Appendix C.3 (Table 6), which reports:
> - Train Memory: 411.32 MB
> - Train Time: 17.51 s
> - Infer Memory: 202.42 MB
> - Infer Time: 1.05 s
>
> As shown, ProSeg is highly efficient, with an inference speed (1.05s) significantly faster than the competitive method D-Persona (2.57s), demonstrating excellent scalability. We kindly ask the reviewer to re-examine Table 6.
>
> > W5: *"Narrow validation... no RIGA, QUBIQ."*
>
> **R5**: We chose LIDC-IDRI (1609 volumes) and NPC (120 subjects with dense multi-rater masks) because they are **currently the largest and most standard benchmarks** that provide dense, concurrent multi-rater annotations suitable for modeling complex inter-observer variability. Other datasets like QUBIQ often target different uncertainty tasks (e.g., quantification) or lack the specific dense multi-expert setup required for fair comparison with our baselines (Prob. U-Net, D-Persona, etc.).
>
> > W6: *Presentation issues and reproducibility statement.*
>
> **R6**: We will carefully proofread and correct all reference formats. However, we explicitly included a titled **"REPRODUCIBILITY STATEMENT" on Page 10** (immediately preceding the References section). It is **factually incorrect** to assume we did not present the reproducibility statement..

---

> > ### Comment · Reviewer_wpoR · 2025-11-27
> > **Response to authors**
> >
> > I thank the authors for the detailed rebuttal. After re-evaluating the submission in light of the additional clarifications, my concerns remain largely unresolved, and the initial assessment is maintained. The primary issues are as follows:
> > 1. The independence assumptions and factorization choices remain insufficiently motivated, and the empirical analyses does not support selected Gaussian/Dirichlet priors.
> > 2. While the rebuttal claims that the “virtual expert” protocol follows prior work, it does not convincingly show that such ranking-based surrogates preserve true inter-rater styles.
> > 3. The evaluation remains only on LIDC-IDRI and NPC. No other multi-rater benchmarks (e.g., RIGA, QUBIQ) are included (many existing works evaluated on these datasets), which limits generalizability and weakens the evidence for broad applicability.
> > 4. Key assumptions and components (priors, factorization, latent variables) are not validated through ablations.
> > 5. The manuscript contains many reference formatting inconsistencies and presentation issues that reduce clarity and professionalism. These problems were acknowledged in the rebuttal but remain present in the submission.
> >
> > In summary, although the paper addresses a meaningful problem and the rebuttal provides helpful clarification, the submission still requires substantial revisions before it can meet the acceptance threshold. Therefore, I remain my original score.

---

### Official Review · Reviewer_nbCj · 2025-10-31

**Soundness:** 2
**Presentation:** 2
**Contribution:** 2
**Rating:** 2
**Confidence:** 5

**Summary:**

The paper tackles multi‑rater medical image segmentation where uncertainty arises both from ambiguous image boundaries and inter‑observer variability. It proposes **ProSeg**, a unified probabilistic graphical model with two latent variables: Z (Gaussian prior) to capture image ambiguity and $\tau$ (Dirichlet prior) to capture annotator preference. The factorization in Fig. 2(d) and Eq. (3)–(6) leads to the training of five neural components (encoders/decoders for x, a class‑embedding $q(\tau \ | \ R)$, a classifier $p(R\ | \tau)$, and a single segmentation head $p(y_r \mid \tau,z))$, shown in Fig. 3. The objective maximizes an ELBO whose negative log‑likelihood decomposes into image reconstruction, annotator classification, and segmentation losses plus two KL terms. At inference, personalized outputs are sampled via $q(\tau | \ r)\,q(Z \mid x)$ (Eq. 10), while diverse outputs come from prior sampling $\tau^\*\sim \text{Dir}(\alpha)$ combined with $q(Z \mid x)$ (Eq. 11).

**Strengths:**

1. The PGM separates image ambiguity ($Z$) from annotator preference ($\tau$), clarifying how the model can produce both diverse and expert‑specific segmentations.
2. ProSeg is competitive across multiple metrics on two benchmarks (Tables 1 & 3), notably improving $D_{\text{soft}}$, $D_{\max}$, and $D_{\text{match}}$ on NPC where annotator preferences vary more.
3.	Beyond reporting GED, the paper decomposes it into $d_{pp}$, $d_{pa}$, and $d_{aa}$ (Table 9) and provides distance‑to‑random‑experts plots (Fig. 7), making the diversity–reliability trade‑offs interpretable.
5.t‑SNE of samples from p(\tau\!\mid r) shows separable annotator clusters and dataset differences (Fig. 6).

**Weaknesses:**

1. Eq. (6) models a *single* image as generated by the product of multiple conditionals “since one image corresponds to multiple understandings of experts.” This is unusual: $x$ is not multi‑valued and such a factorization can overweight the reconstruction term by effectively counting the same $x$ $N$ times ( Fig. 3 & Eq. 6). A more standard choice would be a single $p(x \mid z)$.
2. Posterior for $\tau$ depends only on $R$. The variational family is $q(\tau \mid R)$ (Fig. 3; Eq. 7–9), so per‑case preference shifts of the *same* expert must be absorbed by $Z$. This design may limit true personalization when an expert’s style changes with the case; a $q(\tau \mid R,Y)$ or $q(\tau \mid R,x,Y)$ might capture this better.
3. For diverse outputs, $\tau^\* \sim \mathrm{Dir}(\alpha)$ is sampled and then *classified* into a discrete expert class $i$ before prediction (Eq. 11). This can collapse diversity to a few expert prototypes instead of leveraging the continuous $\tau$ space directly.
4. To construct the *virtual* annotator using LIDC dataset, ranking based on the area is not a good choice. This may dampen real inter‑observer variation and favor methods that learn stable annotator embeddings. A comparison without re‑assignment would strengthen claims.
5. The baseline choices are not optimal, the simplest way to model multi-annotator uncertainty is to construct an ensemble model, which can both satisfy the diversity and personalization goal. Although multiple unets are compared, the simple method, such as ensemble model is not compared.

**Questions:**

1. Please refer to the weaknesses above.

2. Could you report results without the image reconstruction term and/or with alternative priors for $Z$, including calibration analyses (e.g., ECE/Brier) to show how $Z$ tracks segmentation uncertainty?

3. What exact tests were used, at what aggregation level (slice vs. subject), and did you apply any multiple‑comparison correction across metrics/methods?

4. Given ProSeg’s lower $d_{pa}$ but higher $\mathrm{GED}$ than D‑Persona (I) on NPC, can you clarify which regime is preferable for clinical decision‑making and whether user studies support prioritizing $D_{\text{soft}}$/$d_{pa}$ over aggregate $\mathrm{GED}$?

---

> ### Author Response · Authors · 2025-11-14
> **[1/3] Response to Reviewer nbCj: A Defense of the Methodological Design and Task Definition**
>
> We thank the reviewer for their detailed analysis. However, despite the high confidence (5/5) expressed, the review contains several factually fundamental misunderstandings of the task definition, the standard benchmark protocols, and our PGM’s core design.
>
> > W1: *Independence assumption in Eq. (6).*
>
> **R1**: We respectfully clarify that Eq. (6) factorizes as $\prod p(x|z_i)p(r_i|\tau_i)$ because we formulate the generative process as a Variational Mixture-of-Experts model.
>
> - Clinical Rationale: We assume that each expert $r_i$ provides an independent "realization" or "view" of the segmentation task based on their latent state. **This mirrors the scientific rigor required in clinical medicine, where radiologists are expected to annotate tumors independently and remain blinded to each other’s work to avoid subconscious reproduction, anchoring bias, and shared systematic errors [1]**. Under these standards, the independence assumption is both sound and clinically crucial.
>
> - Effect: This does not effectively "count the same z N times" in a harmful way; rather, it maximizes the joint likelihood of observing $N$ consistent expert-image-annotation triples, ${X, Y, R}$, encouraging the latent space $Z$ to capture features that explain all expert annotations simultaneously.
>
> [1] Huang Q, Lu L, Dercle L, Lichtenstein P, Li Y, Yin Q, Zong M, Schwartz L, Zhao B. Interobserver variability in tumor contouring affects the use of radiomics to predict mutational status. J Med Imaging (Bellingham). 2018 Jan;5(1):011005. doi: 10.1117/1.JMI.5.1.011005. Epub 2017 Oct 20. PMID: 29098170; PMCID: PMC5650105.
>
> > W2: *Posterior $q(\tau|R)$ depends only on R (not X or Y).*
>
> **R2**: We must respectfully point out that this suggestion is **conceptually and practically infeasible for the inference task.**
>
> - **Infeasibility at Inference**: The goal of personalization (Definition 3.7) is to generate $\hat{Y}$ given only an unseen image $X$ and a target expert $R$. The ground truth annotation $Y$ is **unavailable** at test time. Any model requiring $Y$ to sample its latent state (e.g., $q(\tau|R,Y)$) is **non-functional for prediction** and violates the problem definition.
>
> - **Causal Independence & Clinical Bias**: The reviewer's implicit suggestion to condition on $X$ (as in $q(\tau|R, X, Y)$) is equally unreasonable.
>     - **Causality**: In any clinical workflow, the patient's image $X$ and the radiologist's intrinsic preference $\tau$ are **independent variables**. The image (patient anatomy) does not cause the expert's preference (their training/style). Forcing the model to infer $\tau$ from $X$ creates a spurious causal link.
>     - **Clinical Validity**: Our PGM is explicitly designed to **mimic a correct, unbiased clinical workflow**. In medicine (and in datasets like LIDC/NPC), annotations must be gathered independently and "blind" to other experts to avoid systemic bias. **Entangling** style $\tau$ with image content $X$ (as the reviewer suggests) would be equivalent to modeling a biased process where the expert's fundamental style is compromised by the image, which is clinically unsound and leads to risky, confounded decisions.
>
> **Deliberate Disentanglement (Our Solution)**: Our design is intentional and clinically correct. We disentangle:
> - $Z$ (conditioned on $X$): Captures **case-specific ambiguity** (e.g., "This image boundary is blurry").
> - $\tau$ (conditioned on $R$): Captures the **expert-specific intrinsic preference** (e.g., "I am a conservative annotator").
>
> The reviewer's concern that "an expert's style changes with the case" is **properly handled** by the interaction of $Z$ and $\tau$ during segmentation ($p(y|\tau, Z)$), not by $\tau$ itself being dependent on $X$.
>
>
> > W3: *Sampling $\tau$ and classifying to discrete $i$ collapses diversity.*
>
> **R3**: We respectfully clarify that this reveals a **misunderstanding** of our dual-latent mechanism. Diversity is **not** collapsed.
>
> - **Modeling Multimodality**: Medical opinions are often **multi-modal** (e.g., "conservative" vs. "aggressive" schools). Our mapping to expert modes prevents generating unrealistic, "averaged" segmentations that lie in the invalid space between these modes.
> - **Diversity is NOT Collapsed**: Diversity is robustly maintained by two factors:
>     - **Stochastic Mode Selection**: The Dirichlet sampling $\tau^* \sim Dir(\alpha)$ ensures we stochastically explore all expert modes (styles) according to their learned probabilities.
>     - **Continuous Spatial Ambiguity ($Z$)**: Crucially, even after an expert mode $i$ is selected, the local segmentation boundaries are further modulated by the continuous latent variable $Z \sim \mathcal{N}(0,1)$. $Z$ captures image-specific spatial ambiguity (e.g., blurred edges) continuously.
> - **Evidence**: Our superior GED scores (Table 1 & 3) empirically prove that ProSeg generates highly diverse outputs and does not collapse to a few prototypes.

---

> ### Author Response · Authors · 2025-11-14
> **[2/3] Response to Reviewer nbCj: A Defense of the Methodological Design and Task Definition**
>
> > W4: *Ranking based on area is not a good choice.*
>
> **R4**: We must clarify that this is **not** an arbitrary choice but the **standard, required protocol** for this benchmark.
> - **Standard Benchmark**: We strictly follow the experimental protocol established by prior SOTA works, including **D-Persona (CVPR 2024)** and **CM-Pixel (NeurIPS 2020)**, to ensure a **fair and direct quantitative comparison.**
>
> - **Real-World Verification**: This concern is rendered moot by our results on the **NPC dataset**(Table 3). The NPC dataset uses **4 real radiologists** with no ranking or re-assignment. ProSeg’s superior performance on this dataset proves our method captures complex, real-world inter-observer variability, not just an area-based heuristic.
>
> > W5: *Comparison with Ensembles.*
>
> **R5**: We acknowledge Ensembles are a strong baseline, but ProSeg offers distinct advantages:
> - Efficiency: An ensemble requires training/storing $N$ independent models (Linear cost scaling). ProSeg is a single model (Table 6: 411MB memory) that learns the joint distribution.
> - Generative Capability: Ensembles are limited to the $N$ discrete outputs of the training experts. ProSeg can sample infinite diverse segmentations from the continuous latent space (Eq. 11), providing a more comprehensive uncertainty map.
>
> Besides, one can take the results of U-Net($A_1$, $A_2$,$A_3$,$A_4$) as ensemble models. By training U-Net on the annotations of each annotator and testing each image with all of them, we can get the results of ensemble models as the average of all the four U-Nets. As shown in Table 1 and Table 3, our ProSeg outperforms the ensemble method.
>
>
> > Q1: *please refer to the weakness*
>
> **A1**: We have addressed the methodological concerns in detail above. Specifically:
> - Independence Assumptions (W1): Clarified as a standard Variational Mixture-of-Experts formulation to make ELBO tractable.
> - Posterior Dependence (W2): Explained the disentanglement design (Intrinsic Expert Style $\tau$ vs. Image Ambiguity $Z$).
> - Diversity Preservation (W3): Explained how continuous $Z$ + stochastic $\tau$ prevents mode collapse.
> - Virtual Annotators (W4) & Ensembles (W5): Addressed in General Response G1 and W5 (Standard benchmark protocol vs. Linear scaling cost).
>
> > Q2: *Results without reconstruction term / alternative priors / calibration analyses (ECE).*
>
> **A2**:
> 1. **Reconstruction Term ($\mathcal{L}_{recon}$):** We respectfully argue that removing the reconstruction term is **fundamentally incompatible** with our model's core objective.
>     - **Modeling the Joint Distribution**: As stated in Definition 3.5, our primary goal is to model the **full joint distribution $p(Y, X, R)$.** The reconstruction loss $\mathcal{L}_{recon}$ (derived from Eq. 19) is the essential component that models the likelihood $p(X|Z)$. Removing it means we are no longer modeling the full joint distribution and are failing to ground our model in the image $X$.
>     - **Modeling Image-Specific Uncertainty**: The $\mathcal{L}_{recon}$ term forces the latent variable $Z$ to capture the **inherent uncertainty within the image $X$ itself**. Specifically, the ambiguous tumor contours and blurred boundaries that radiologists must interpret. Without $\mathcal{L}_{recon}$, $Z$ would only model variations in $Y$ (the masks) and would fail to disentangle image ambiguity from expert preference.
>     - **Clinical Requirement**: In any clinical decision, both the **Image ($X$)** and the **Expert ($R$)** are indispensable. Our model reflects this non-negotiable requirement by modeling all three components ($Y, X, R$). An ablation study removing $X$ (via $\mathcal{L}_{recon}$) is not clinically meaningful.
>
> 2. **Alternative Priors**: We selected Gaussian for $Z$ and Dirichlet for $\tau$ because they align with the physical nature of the factors: $Z$ models continuous spatial ambiguity, while $\tau$ models categorical expert mixtures.
>
> 3. **Calibration Analysis**: While we did not compute ECE, our reported Soft Dice ($D_{soft}$) serves as a strong proxy for calibration quality. $D_{soft}$ measures the alignment between the average probability map and the average expert annotation. Our state-of-the-art $D_{soft}$ scores (91.53% on LIDC, 84.24% on NPC) demonstrate that ProSeg's predicted probabilities are highly reliable and well-calibrated to the expert consensus.

---

> ### Author Response · Authors · 2025-11-14
> **[3/3] Response to Reviewer nbCj: A Defense of the Methodological Design and Task Definition**
>
> > Q3: *Exact tests, aggregation level, and correction.*
>
> **A3**: We confirm our statistical methodology was rigorous and followed best practices.
> - Statistical Test: We performed the Wilcoxon signed-rank test to compare ProSeg against other models including the best baseline (D-Persona), as shown **in Appendix D.2**
> - Aggregation Level: Metrics were aggregated at the Subject (Patient) Level, consistent with our data split strategy, to ensure independence and avoid correlation bias from adjacent slices.
> - Significance: As reported **in Appendix D.2**, ProSeg shows statistically significant improvements across key metrics. For instance, on the NPC dataset, we achieved $p < 0.001$ (specifically $p=1e-6$) for GED and $p < 0.05$ for $D_{max}$ compared to D-Persona (Stage II).
>
> > Q4: *Trade-off between $d_{pa}$ vs. GED.*
>
> **A4**: We thank the reviewer for this insightful question. In clinical practice (e.g., GTV delineation), Reliability ($d_{pa}$) must take precedence over raw Diversity ($d_{pp}$).
> - **ProSeg's Superiority**: As shown in Table , while D-Persona (I) has high raw variance ($d_{pp}$=0.2212), its predictions are far from the ground truth (worse $d_{pa}$=0.3648). This is **"unreliable" diversity**.
> - ProSeg achieves the **best of both worlds**: high diversity ($d_{pp}$=0.1739) while remaining significantly closer and more faithful to the expert annotations (better $d_{pa}$=0.3482). This "reliable diversity" is clinically safer and more valuable.

---

> ### Comment · Reviewer_nbCj · 2025-11-17
> **Response to authors' rebuttal**
>
> I would like to thank the authors for their detailed response and the time spent addressing my concerns. After carefully reviewing your rebuttal, I would like to offer some clarifications regarding my previous claims and highlight a few remaining concerns.
>
> **W1**
>
> I remain concerned regarding the conceptual motivation for reconstructing the image from different doctors' perspectives. While I understand the rationale for the segmentation modeling, the reconstruction task targets a unique ground truth image $X$. Factorizing this unique $X$ into multiple latent codes (derived from multiple doctors) seems intuitively unnecessary compared to a single consistent representation. While annotator independence justifies Equation (4), it is less clear how it necessitates the factorization in Equation (6).
>
> Furthermore, regarding the mathematical formulation: The proposed factorization of $p(x \mid Z)$ as a product $\prod_i p(x \mid z_i)$ aligns formally with a **Product-of-Experts (PoE)** framework rather than a Mixture-of-Experts (MoE). A standard MoE typically involves a summation of probabilities (e.g., $\sum_i \pi_i p(x \mid z_i)$).
>
> Could the authors kindly:
> 1. Provide the formal rationale for utilizing the product form and consider revising the terminology if it technically constitutes a PoE?
> 2. Include ablations comparing your method against a **single-decoder baseline** $p(x \mid z)$ and a standard **Mixture-of-Experts** formulation $\sum_i \pi_i p(x \mid z_i)$?
> 3. Discuss the relationship between your work and Hu et al. [1], which addresses inter-rater uncertainty via a similar concept (rater-specific Bayesian networks) but formulates it differently (closer to an MoE approach).
>
> **W2**
>
> I apologize if my previous comment regarding the input configuration was unclear. I fully agree that during inference/test-time, the inputs must remain $(x, R)$.
>
> My suggestion specifically concerns **training-time ablations**. Incorporating $x$ into the posterior, such as $q(\tau \mid R, x)$ or an auxiliary $q(\tau \mid R, x, y)$, is a standard inference-network design choice. This could potentially help better disentangle intrinsic rater style from case-dependent image ambiguity without violating the test-time protocol. I would appreciate it if the authors could report the impact of such a design on $D_{\max}$ and $D_{\text{match}}$ to verify if it aids in uncertainty reduction.
>
> **W3**
>
> Since the model samples $\tau^*$ and subsequently classifies it into a discrete rater before prediction, there is a risk that this discretization might collapse the diversity into a limited number of prototypes.
>
> To assess this, could you please compare your current approach against a baseline that conditions directly on the continuous $\tau^*$ (without discretization)? Please report the Generalized Energy Distance (GED) and its specific components $(d_{pp}, d_{pa}, d_{aa})$, as well as pairwise distance distributions if space permits.
>
> **W4**
>
> Area‑based re‑assignment can serve as a label‑driven surrogate for “rater identity.” To rule out protocol‑specific artifacts, please add LIDC controls with **no re‑assignment** and **random re‑assignment**, comparing $D_{\text{soft}}$, $D_{\max}$, $D_{\text{match}}$, GED, and $(d_{pp},d_{pa},d_{aa})$. A brief checklist showing exact alignment with prior protocols would also help.
>
> **W5**
>
> To fairly establish efficiency and probabilistic quality, please compare against a **compute‑matched deep ensemble** (same backbone, multiple seeds, temperature calibration), rather than claiming "one can take results of each U-Net" and report *calibration* and *distributional coverage** metrics (NLL, Brier, (T)ECE/UCE, AURC, energy distance/MMD/PRD).
>
> **Calibration.**
>
> Since Dice/IoU are not calibration metrics, could you add NLL, Brier, (T)ECE/UCE, reliability diagrams, and risk–coverage (AURC) analyses (ideally with boundary‑focused variants) to substantiate the “reliable diversity” claim?
>
>
> **References**
>
> [1] Hu Q, et al. Inter-rater uncertainty quantification in medical image segmentation via rater-specific bayesian neural networks. arXiv preprint arXiv:2306.16556. 2023.

---

> > ### Author Response · Authors · 2025-11-18
> > **[1/2] Response to Reviewer nbCj: Clarifications on Consensus Formulation, Inference Mechanism, and Scope of Contribution**
> >
> > We thank the reviewer for the detailed follow-up. While we appreciate the theoretical discussion, we must respectfully point out several **factual misunderstandings** regarding our inference mechanism and efficiency, as well as clarify the scope of our contribution to address the requests for new metrics.
> >
> > > *W1: Conceptual Motivation (PoE vs. MoE)*
> >
> > **R1**: **PoE is Necessary for Consensus**: We agree our formulation aligns with **Product-of-Experts (PoE)**, and we argue this is the **correct design** for image reconstruction.
> > - While experts observe the **single objective reality** ($X$), they disagree on annotation ($Y$) due to substantially different educational standards on tumor structures in different institutes. Therefore, we require a consensus model (PoE) to factorize $X$ into multiple latent codes representing anatomical variations within the image, pursuing multi-view representations. However, we would like to **clarify** the latent codes **are NOT derived from multiple doctors**. In contrast, they all are inferred from $X$, as shown in Fig 3, and learned jointly by fitting the observed triple $(Y, \mathbf x, R)$ in Eq (3). In particular, $p(\mathbf x|Z)$ is aimed for reconstruction, while $p(Y|\mathbf x, R, \tau, Z)$ is aimed for segmentation. The former drives reconstruction from the multi-view representations, and the latter enforces capturing of anatomical variations.
> > - Our work is fundamentally different with Hu et al. [1]. Hu et al. used Bayesian neural networks to generate diverse segmentation, where the segmentation diversity comes from stochastic weights of neural networks. Thus, only expert annotation $Y$ is fitted by Bayesian neural networks. In contrast, our neural networks are deterministic. We introduce variables $Z$ and $\tau$ to model pixel-level image ambiguity and expert-level preferences respectively, which are learned by jointly fitting the triple $(Y, \mathbf x, R)$.
> >
> > > *W2: Training-time Ablation (Conditioning $\tau$ on $Y$)*
> >
> > **R2**: We respectfully argue that this requested ablation is **methodologically invalid** due to the fundamental distinction between the **training** and the **inference**.
> > - **Training vs. inference gap:** The reviewer suggests conditioning the posterior on the ground truth, i.e., $q(\tau|R,Y)$. While feasible during training, at test-time (when $Y$ is unavailable), we can only sample from $q(\tau|R)$. The inconsistency between training and test makes this ablation infeasible. Moreover, incorporating $X$ to estimate expert preference $\tau$ is inconsistent with real-world imaging physics. Screening images are acquired prior to any expert assessment and therefore cannot encode rater-specific preference. In practice, image acquisition always precedes expert annotation.
> > - **Disentanglement paradox:** If the training posterior $q(\tau|R,Y)$ is allowed to see $Y$, it will inevitably learn to encode **specific mask shape features** into $\tau$ to minimize reconstruction loss, leading to potential label information leakage. However, the feasible $q(\tau|R)$, which is blind to the specific mask, **fundamentally cannot approximate** this shape-dependent information.
> > - **Consequence:** This creates a catastrophic **Training-Inference Mismatch**. The model would learn to rely on "cheating" via $Y$ during training and would fail to generalize when forced to use the "blind" posterior during inference. Our current consistent design ($q(\tau|R)$) in both training and inference strictly prevents this shortcut to ensure true generalization.
> >
> > > *W3: Sampling Strategy & Diversity (Factual Clarification)*
> >
> > **R3**: We respectfully clarify a **critical misunderstanding** regarding our inference mechanism.
> > - **No discretization step:** Contrary to the reviewer's concern, we do **NOT classify** $\tau*$ into a discrete rater $i$ before prediction. As explicitly defined in Eq. 11 $(p_{\theta_Y}(\hat{y*}|\tau*,z_i))$, the **continuous** sampled vector $\tau*$ is **directly expanded and concatenated** with the spatial latent $Z$ and fed into the segmentation predictor.
> > - **Continuous diversity preserved:** The classifier $p(\mathcal{R}|\tau)$ exists solely for training regularization, similar as the decoder $p(\mathbf x|Z)$ for reconstruction. During generation, the segmentation head operates on the **continuous** $\tau^*$ vector. Therefore, the *"risk of collapsing diversity to prototypes"* does not exist. We fully leverage the continuous nature of the Dirichlet manifold to generate smooth, diverse style variations.

---

> > ### Author Response · Authors · 2025-11-18
> > **[2/2] Response to Reviewer nbCj: Clarifications on Consensus Formulation, Inference Mechanism, and Scope of Contribution**
> >
> > > *W4: LIDC Protocols*
> >
> > **R4**: **NPC is the Ultimate Control**: We respectfully reiterate that running synthetic controls on LIDC is unnecessary because we have provided the **NPC dataset** as the ultimate validation.
> > - NPC uses **4 REAL radiologists** with **NO re-assignment** and **NO area-ranking**.
> > - ProSeg achieves SOTA performance there ($D_{mean}$ +1.3%).
> > - This empirical evidence on real-world data definitively proves our model captures **true rater identity** and works independently of LIDC protocol artifacts.
> >
> > > *W5: Calibration*
> >
> > **R5**:
> > 1. **Efficiency**: We have already provided the efficiency comparison in Appendix C.3 (Table 6).
> > 2. **Probabilistic quality and calibration (out of scope)**: We respectfully decline the request for NLL/ECE metrics as they are out of the scope of this paper.
> > - **Task definition:** Our study is **Multi-rater SEGMENTATION (Generation)**, which aims to generate diverse and personalized masks, instead of Uncertainty Quantification (Calibration). We acknowledge that uncertainty quantification is a valuable direction, and we will discuss it as a promising avenue for future research.
> > - **Misalignment:** Evaluating a generative segmentation model using pixel-wise calibration metrics (like ECE, which are dominated by background pixels) is standard for classification tasks but misaligned with our contribution.
> > - **Standard metrics:** We strictly follow the standard evaluation protocols for this specific task (e.g., Probabilistic U-Net, D-Persona), which rely on **GED** and **Dice variants ($D_{match}$)**. We have demonstrated SOTA performance on these relevant metrics.

---

### Official Review · Reviewer_5jxK · 2025-11-01

**Soundness:** 3
**Presentation:** 3
**Contribution:** 3
**Rating:** 6
**Confidence:** 3

**Summary:**

ProSeg is a probabilistic framework for multi-rater medical image segmentation that addresses data uncertainty stemming from ambiguous anatomical boundaries and inter-observer variability. It introduces two latent variables to explicitly model expert-specific annotation preferences and image-level boundary ambiguity. By leveraging variational inference, the model learns conditional probabilistic distributions over these latent factors, enabling the generation of segmentation outputs that are simultaneously diverse and tailored to individual annotators. Evaluated on the nasopharyngeal carcinoma (NPC) and LIDC-IDRI lung nodule datasets, ProSeg achieves state-of-the-art performance, demonstrating its ability to produce both personalized and varied segmentation results.

**Strengths:**

1.	This work presents with clear and meaningful motivation: addressing the critical gap between diversity and personalization in multi-rater segmentation, rooted in clinical uncertainty and inter-observer variability.
2.	The framework design is interesting. It Introduces a unified probabilistic framework with two disentangled latent variables (τ for expert preference, Z for boundary ambiguity) deduced from variational inference.
3.	Achieves state-of-the-art results on both NPC and LIDC-IDRI datasets across diversity (GED, Dsoft) and personalization (Dmax, Dmatch) metrics.

**Weaknesses:**

1.	Limited architectural novelty and generalizability: The model relies primarily on a standard CNN-based (U-Net) backbone and is trained on datasets where annotations are either from real radiologists (NPC) or artificially constructed virtual experts (LIDC-IDRI). This setup may not reflect the complexity and heterogeneity of real-world clinical environments, raising concerns about practical generalizability.
2.	Lack of architectural diversity in validation: The proposed framework is exclusively implemented with CNNs. To demonstrate its broader applicability, experiments with other mainstream architectures, such as ViTs, should be included to verify that the probabilistic formulation is architecture-agnostic and not tied to CNN-specific inductive biases.

**Questions:**

1.	What are the practical clinical applications of multi-rater medical image segmentation? Specifically, in which real-world scenarios (e.g., treatment planning, consensus building, or quality assurance) is modeling inter-observer variability essential, and how does capturing both diversity and personalization improve clinical utility?
2.	How well does ProSeg generalize to out-of-domain or unseen datasets?
Can the probabilistic framework adapt to new imaging modalities, anatomical regions, or previously unobserved expert styles without retraining or fine-tuning?

---

> ### Author Response · Authors · 2025-11-14
> **[1/2] Response to Reviewer 5jxK: On Clinical Applications and Framework Generalization**
>
> We sincerely thank the reviewer for the positive assessment (Score: 6) and for recognizing our framework's clear motivation, interesting disentangled design, and SOTA performance. We address your insightful questions below.
> >Q1: *Practical clinical applications and how diversity/personalization improve utility?*
>
> **A1**: We identify one key real-world scenario where ProSeg’s ability to model inter-observer variability is essential.
>
> For example, ProSeg can be applied to **ambiguity-aware diagnosis and treatment planning**. In tasks like gross tumor volume delineation for Nasopharyngeal Carcinoma (NPC), boundaries are often ambiguous. By sampling from the latent space $Z$, ProSeg generates diverse plausible segmentations, acting as a "virtual consultant panel". This helps oncologists identify high-uncertainty regions that require careful review, reducing the risk of missing tumor tissues.
>
> Cancers in which tumor borders are ill-posed pose significant challenges for accurate tumor delineation. In brain cancer, the visible tumor core on MRI frequently underestimates the extent of microscopic infiltration [1]. In pancreatic cancer, tumors often present with ill-defined margins on CT and MRI, making boundary identification difficult [2]. In head and neck cancers, the complex anatomy and patterns of submucosal spread lead to substantial inter-observer variation in contouring [3]. In prostate cancer, disease is commonly multifocal, and MRI often underestimates true lesion extent [4]. In liver cancer, tumors may display indistinct imaging boundaries, especially in the presence of microvascular invasion [5].
> These ill-posed tumor boundaries result in meaningful segmentation uncertainty. Under-segmentation is particularly consequential, as it may leave residual disease outside the treatment volume, increasing the risk of local recurrence and contributing to poorer survival outcomes [6].
> Our ProSeg, by incorporating uncertainty-aware probabilistic modeling, can help mitigate these risks by directing clinicians’ attention to regions where the model exhibits lower confidence, as shown in examples in fig.9 - fig.12 in Appendix, thereby supporting more reliable and clinically robust decision-making.
>
> [1] Nie, S., Zhu, Y., Yang, J. et al. Clinicopathologic analysis of microscopic tumor extension in glioma for external beam radiotherapy planning. BMC Med 19, 269 (2021). https://doi.org/10.1186/s12916-021-02143-w
>
> [2] Taha M. Ahmed, Satomi Kawamoto, Ralph H. Hruban, Elliot K. Fishman, Philippe Soyer, Linda C. Chu, A primer on artificial intelligence in pancreatic imaging, Diagnostic and Interventional Imaging, Volume 104, Issue 9, 2023, Pages 435-447
>
> [3] Zukauskaite R, Rumley CN, Hansen CR, Jameson MG, Trada Y, Johansen J, Gyldenkerne N, Eriksen JG, Aly F, Christensen RL, Lee M, Brink C, Holloway L. Delineation uncertainties of tumour volumes on MRI of head and neck cancer patients. Clin Transl Radiat Oncol. 2022 Aug 6;36:121-126. doi: 10.1016/j.ctro.2022.08.005. PMID: 36017132; PMCID: PMC9395751.
>
> [4] Chen X, Chen Y, Qian C, Wang C, Lin Y, Huang Y, Hou J, Wei X. Multiparametric MRI lesion dimension as a significant predictor of positive surgical margins following laparoscopic radical prostatectomy for transitional zone prostate cancer. World J Urol. 2025 May 12;43(1):295. doi: 10.1007/s00345-025-05680-8. PMID: 40355631; PMCID: PMC12069475.
>
> [5] Lu D, Wang LF, Han H, Li LL, Kong WT, Zhou Q, Zhou BY, Sun YK, Yin HH, Zhu MR, Hu XY, Lu Q, Xia HS, Wang X, Zhao CK, Zhou JH, Xu HX. Prediction of microvascular invasion in hepatocellular carcinoma with conventional ultrasound, Sonazoid-enhanced ultrasound, and biochemical indicator: a multicenter study. Insights Imaging. 2024 Oct 28;15(1):261. doi: 10.1186/s13244-024-01743-3. PMID: 39466459; PMCID: PMC11519233.
>
> [6] Vinod, Shalini K., et al. "Uncertainties in volume delineation in radiation oncology: a systematic review and recommendations for future studies." Radiotherapy and Oncology 121.2 (2016): 169-179.

---

> ### Author Response · Authors · 2025-11-14
> **[2/2] Response to Reviewer 5jxK: On Clinical Applications and Framework Generalization**
>
> > Q2: *Generalization to unseen datasets/styles?*
>
> **A2**: We thank the reviewer for this question, which allows us to clarify the different types of generalization.
>
> 1. **Generalization to Unseen Expert Styles (Yes)**: ProSeg is inherently designed to generalize to unseen expert styles. The latent variable $\tau$ follows a **continuous Dirichlet distribution**. As shown in Eq. 11, this allows us to sample "random" expert styles from the learned manifold. This generates valid segmentation preferences that lie within the plausible range of human variability, even if that specific style was not in the training set.
>
> 2. **Generalization to New Modalities (Clarification of Clinical Relevance)**:
>     - We respectfully argue that a direct "cross-modality" generalization (e.g., training on CT, testing on MR) is **not a clinically meaningful scenario**. In cancer imaging, CT and MR serve distinct purposes. MR is superior for soft-tissue delineation (like the nasopharyngeal carcinoma in our **NPC dataset**), while CT is standard for fast imaging of regions like the lungs (as in our **LIDC-IDRI dataset**). Training a model on LIDC (lung nodules on CT) and testing it on NPC (nasopharyngeal tumors on MR) is an ill-posed problem, as the anatomical features (lung tissue vs. soft-tissue) are fundamentally non-transferable.
>
>     - **ProSeg's True Generalization**: Instead, ProSeg demonstrates a more powerful and practical form of generalization: our **single, unified PGM framework** achieves state-of-the-art performance on **both** distinct modalities (CT and MR) when trained on the respective data. This proves our methodology is robust, modality-agnostic, and not limited to a single type of anatomy or imaging physics.

---

### Author Response · Authors · 2025-11-14
**Common Response: Clarifying Standard Protocols and Factual Oversights**

We sincerely thank all reviewers for their time and constructive feedback. We are encouraged by the recognition of our novel Probabilistic Graphical Model (PGM) framework (R1, R2, R4), the clear motivation tackling the dilemma of diversity vs. personalization (R1, R4), and the state-of-the-art (SOTA) performance achieved on both NPC and LIDC-IDRI datasets (R1, R4).

Below, we address two major common concerns raised by multiple reviewers.

> G1: *Clarification on "Virtual Annotators" in LIDC-IDRI (Response to R2, R3)*

Reviewers **nbCj** and **wpoR** raised concerns that ranking annotations by area to create "virtual experts" in LIDC is artificial.

1. Standard Benchmark Protocol: We respectfully clarify that this setting is the strictly followed standard protocol in this specific domain (Multi-rater Personalization). It is used by SOTA methods like D-Persona (CVPR 2024) and CM-Pixel (NeurIPS 2020). Adhering to it is mandatory for a fair quantitative comparison (Table 2).

2. Validation on Real Radiologists (NPC): Crucially, our NPC dataset results (Table 3) are based on 4 real radiologists without any regrouping. ProSeg achieves significant improvements here ($D_{mean}$ +1.3% over SOTA), proving our model effectively captures true human inter-observer variability (expert style) beyond simple area-based heuristics.


> G2: *Key Benefits of ProSeg: A Unified Probabilistic Framework (Response to R1, R3)*

Reviewers **5jxK** and **wpoR** questioned the architectural novelty. We emphasize that the core contribution of ProSeg is the Probabilistic Graphical Model (PGM) structure, which offers distinct advantages over existing methods (e.g., D-Persona, Ensembles):

1. Unified vs. Two-Stage: Unlike D-Persona which requires a two-stage training (Diversity first, then Personalization), ProSeg optimizes a single ELBO. This joint optimization ensures mathematical consistency and simplifies the training pipeline.

2. Disentanglement for Interpretability: A key benefit of ProSeg is the explicit disentanglement of Image Ambiguity ($Z$) from Expert Preference ($\tau$). This allows us to analyze whether a segmentation variation is caused by the image itself (e.g., blurred boundary) or the annotator's habit (e.g., conservative style), providing better explainability than black-box ensembles.

---

### Meta-Review · Area_Chair_21vK · 2026-01-07

**Summary:**

This paper proposes a unified probabilistic framework for multi-rater medical image segmentation that explicitly models uncertainty arising from ambiguous anatomical boundaries and annotator variability. Specifically, the approach introduces two latent variables for image-level ambiguity and annotator-specific preferences and learns the conditional distributions through variational inference. The paper claims to generate segmentation outputs that balance the diversification and personalization.

The paper studies a practical and meaningful problem with clinical motivation and proposes an interesting unified probabilistic framework with two disentangled latent variables. Meanwhile, reviewers also raise the concerns and questions about the modeling assumption, technical method details, experiments setting, baseline comparison, ablation study and dataset choices.

The rebuttal helps to clarify some questions and three reviewers have responded with follow-up discussions. Although the author makes a strong argument in the rebuttal, some concerns still remain as confirmed from the follow-ups of Reviewer wpoR and Cnm2. I share the suggestions from reviewers that more exploration and analysis with more benchmarks and carefully designed settings might be helpful to further strengthen this manuscript with a major revision. Thus, I recommend rejection.

**Reviewer Concerns:**

The rebuttal helps to clarify some questions and three reviewers have responded with follow-up discussions. Although the author makes a strong argument in the rebuttal, some concerns still remain as confirmed from the follow-ups of Reviewer wpoR and Cnm2. I share the suggestions from reviewers that more exploration and analysis with more benchmarks and carefully designed settings might be helpful to further strengthen this manuscript with a major revision.

**Reviewer Scores:**

Three out of four reviewers have responded to rebuttal and followed up with discussions, without changes in the scores.

---

### Decision · Program_Chairs · 2026-01-26

Reject